# Rapid and continuous regulating adhesion strength by mechanical micro-vibration

Langquan Shui [1,8], Laibing Jia[2,3,8], Hangbo Li[3], Jiaojiao Guo[4], Ziyu Guo[4], Yilun Liu[5], Ze Liu[1✉] & Xi Chen[6,7✉]

Controlled tuning of interface adhesion is crucial to a broad range of applications, such as space technology, micro-fabrication, flexible electronics, robotics, and bio-integrated devices. Here, we show a robust and predictable method to continuously regulate interface adhesion by exciting the mechanical micro-vibration in the adhesive system perpendicular to the contact plane. An analytic model reveals the underlying mechanism of adhesion hysteresis and dynamic instability. For a typical PDMS-glass adhesion system, the apparent adhesion strength can be enhanced by 77 times or weakened to 0. Notably, the resulting adhesion switching timescale is comparable to that of geckos (15 ms), and such rapid adhesion switching can be repeated for more than $2 \times 10^7$ vibration cycles without any noticeable degradation in the adhesion performance. Our method is independent of surface micro-structures and does not require a preload, representing a simple and practical way to design and control surface adhesion in relevant applications.

[1] Department of Engineering Mechanics, School of Civil Engineering, Wuhan University, 430072 Wuhan, People's Republic of China. [2] Department of Naval Architecture, Ocean and Marine Engineering, University of Strathclyde, G4 0LZ Glasgow, UK. [3] School of Marine Science and Technology, Northwestern Polytechnical University, 710072 Xi'an, People's Republic of China. [4] Department of Engineering Mechanics, School of Mechanics, Civil Engineering and Architecture, Northwestern Polytechnical University, 710072 Xi'an, People's Republic of China. [5] State Key Laboratory for Strength and Vibration of Mechanical Structure, School of Aerospace Engineering, Xi'an Jiaotong University, 710048 Xi'an, People's Republic of China. [6] Department of Earth and Environmental Engineering, Earth Engineering Center, Center for Advanced Materials for Energy and Environment, Columbia University, New York 10027 NY, USA. [7] School of Chemical Engineering, Northwest University, 710069 Xi'an, People's Republic of China. [8] These authors contributed equally: Langquan Shui, Laibing Jia ✉email: ze.liu@whu.edu.cn; xichen@columbia.edu

Natural long-term evolution has enabled some living things to exhibit excellent surface adhesion properties[1–3]. For example, geckos use hierarchical structures in their feet to quickly regulate surface adhesion, and the adhesion/detachment switching time is ~15 ms[4], which allows them to climb walls at a speed as high as 0.8 m/s[5]. The excellent functional properties of biological systems have inspired many studies on the design and fabrication of biomimetic artificial surfaces[1,2,6–16], which show fascinating applications in modern industrial technologies, such as space technology, micro-fabrication technology, flexible electronics, robotics, and bio-integrated devices[7,17,18].

In the past decades, significant progress on improving the adhesion performance (e.g., adhesion strength, adhesion switching, reversibility, and durability) of biomimetic artificial surfaces have been achieved, but at the cost of complexity, versatility, and cost[6]. For example, artificial hierarchical adhesion systems have been reported to possess adhesion strengths as high as 200 kPa[19], higher than that observed in nature (e.g., 100 kPa for a gecko[1,8]), but they usually require complex surface patterning techniques and high adhesion-to-preload ratios[6,9]. Besides, the surface microstructures are susceptible to environmental and intrinsic surface forces, which limits the durability. Considering that the apparent adhesion strength can be influenced by the real contact area, the contact region size[17], and the adhesion work[20], adhesion switching has been proposed by either controlling the load path and failure mode of the interface or constructing a "smart" interface such as a phase transition interface[6,20]. The adhesion switching triggers, such as mechanical[12,15,21–30], electro/magnetic[14,16,31], light[32–34], fluid[35–38], and thermal[39–44] stimulations, have been suggested. However, at present, there have been only a few attempts to provide promising methods to tune the adhesion strength continuously and rapidly, for example, gecko-inspired directional adhesion[12,13,15,16,45–50] and debonding/peeling speed-regulated adhesion[24–30].

Here, we report a very simple but effective technique to robustly regulate surface adhesion to a desired (either strengthened or weakened) strength based on the modulation of normal adhesion by micro-vibration. Combining the experiments with the theoretical model, we found that the adhesion can be either strengthened or weakened by controlling the micro-vibration. Moreover, we observed that the adhesion strength can be maintained at any desired value with good durability and reversibility within the theoretically permissible regulation ranges, while the adhesion switching can be very quick, on the order of $10^1$ ms, which is comparable to that of geckos (15 ms[4]).

The experimental setup is shown in Fig. 1, where a stiff glass sphere is aligned in normal contact to a flat polydimethylsiloxane (PDMS) platform. Micro-vibration is carried out by fixing the PDMS platform to a vibration exciter (two loudspeakers are used in our experiments). During the adhesion test, the stiff contactor is precisely controlled to move along the normal direction of the PDMS platform, while the pulling force exerted on the stiff contactor, the size of the adhesion region, and the displacements of the platform and contactor are monitored in situ by a force sensor, a group of laser displacement sensors, and a high-speed camera, respectively (Fig. 1, see the Methods section). Notably, the lateral motion of the stiff contactor should be prevented during the adhesion test.

## Results

**Adhesion behavior under vibration.** Typical experimental results are shown in Fig. 2. By exciting the PDMS platform (i.e., using the strategy on the left side of Fig. 1b) with a micro-vibration (with a frequency of $\omega/(2\pi) = 450$ Hz and amplitude of $A_b = 68$ μm), we found that the measured pull-off force ($F_{off} =$

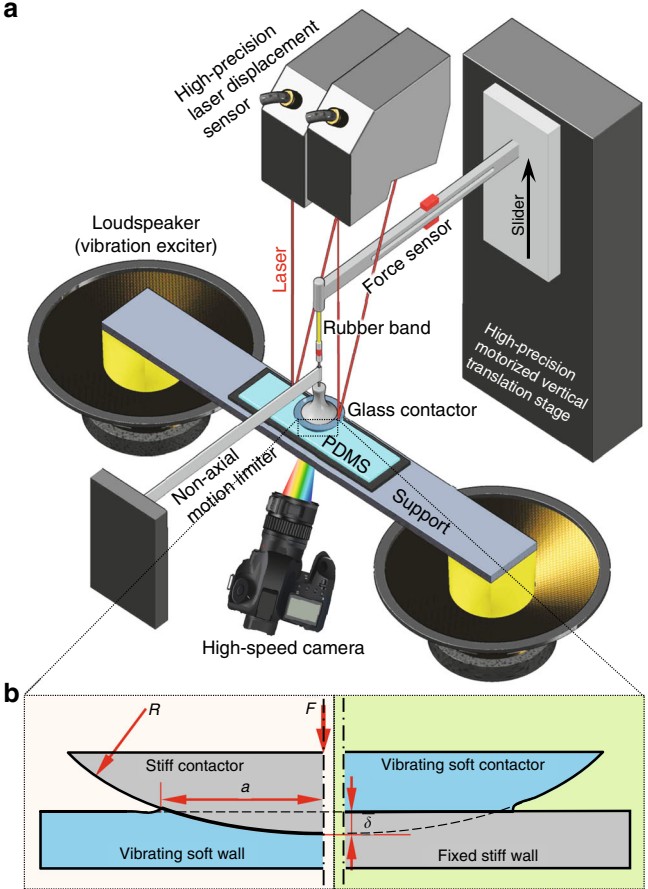

**Fig. 1 Model description. a** Schematics of the experimental setup for the adhesion strength regulation test (see the Methods section). **b** The corresponding mechanical model, where the curvature radius of the contactor, the contact radius, and the penetration depth are $R$, $a$ ($\geq 0$), and $\delta$ (which can be negative), respectively. The contact deformation is maintained by an external force, that is, the contact load, $F$. Under the assumption of small strain, the adhesive contact between a stiff sphere and a vibrating soft substrate (left of **b**) is mechanically equivalent to that between a vibrating soft sphere and a fixed stiff substrate (right of **b**; see Supplementary Fig. 3 and Supplementary Movies 1 and 2).

0.7492 N) considerably increases to more than 75 times of that without micro-vibration ($F_{off,0}$, measured as 0.0097 N) (Fig. 2a). Meanwhile, we observed that during the tests, the change in the contact radius ($a$) within one vibration period is very small. For example, even if the apparent (or average) contact load $\bar{F} \approx 0.6$ N (note that the average tension of the rubber band is equal to $\bar{F}$, Eq. (1)) is ~60 times $F_{off,0}$ (marked by "x" in Fig. 2a), the change in the contact radius is <2% (Fig. 2b). This observation indicates that the modulation of the interface adhesion strength by micro-vibration originates from the dynamic response of the system rather than the change in the effective contact area. In addition, in a durability test in which the apparent contact load $\bar{F}$ was maintained at ~70 times $F_{off,0}$ (marked by "o" in Fig. 2a), we observed that such a considerable regulation of adhesion strength can be robustly repeated over $2 \times 10^7$ vibration cycles (Fig. 2c). Interestingly, we observed that the contact radius changes asymmetrically within one cycle of micro-vibration (Fig. 2b), which could originate from the effect of the rate of change in the adhesion energy, bulk viscoelasticity, inertia, and so on, reflected in the different crack growing (with $a$ decreasing) and healing (with $a$ increasing) rates. It is noted that the above adhesion

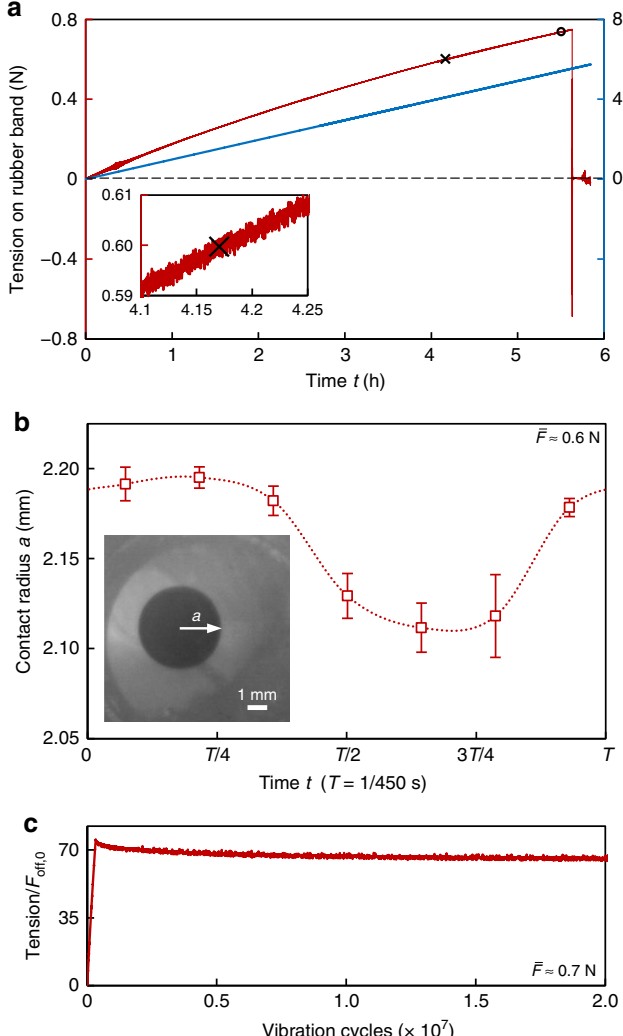

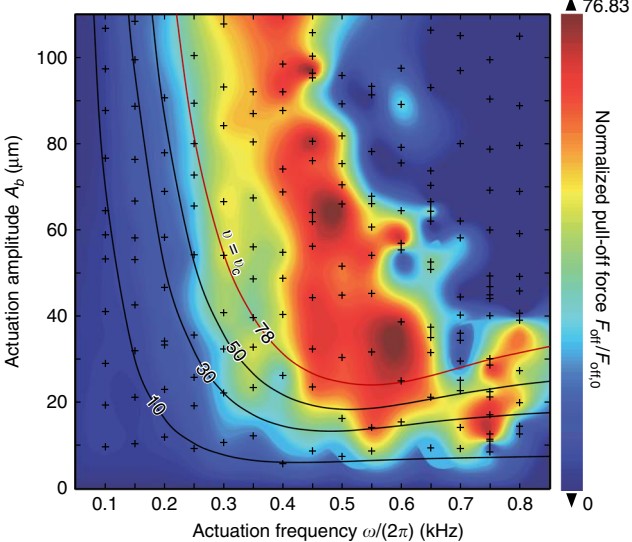

**Fig. 3 Measured normalized pull-off force $F_{off}/F_{off,0}$ with different actuation frequencies and amplitudes.** The experimental data are marked by the plus sign. In this plot, $F_{off}/F_{off,0}$ ranges from 0 to 76.83. Four contours ($F_{off}/F_{off,0} = 10, 30, 50, 78$) are plotted based on theoretical calculations [using Eq. (3), for which the parameters are the same as those in Fig. 5].

**Fig. 2 Typical experimental results for regulating the adhesion strength.** The frequency $\omega/(2\pi) = 450$ Hz. The amplitude $A_b = 68$ μm. **a** The measured pulling force in the rubber (red line) and the upward displacement of the slider (blue line) as a function of time. **b** The change in contact radius with time within one vibration period under the condition of $\bar{F} \approx 0.6$ N (point "x" in **a**), which is ~60 times the pull-off force without micro-vibration ($F_{off,0}$, measured as 0.0097 N). The data (here denoted as $a_{ij}$) are collected from three consecutive vibration cycles in one trial (correspond to $i = 1$–3), and each cycle contains 7 data points (correspond to $j = 1$–7). The dotted curve connects the 7 average values $\bar{a}_{\cdot j} = \sum_{i=1}^{n} a_{ij}/n$, $n = 3$. Each bar represents the standard errors (SE, calculated by $SE_{\cdot j} = \sqrt{\sum_{i=1}^{n} (a_{ij} - \bar{a}_{\cdot j})^2 / [n(n-1)]}$, $n = 3$), and demonstrates the interval $[\bar{a}_{\cdot j} - SE_{\cdot j}, \bar{a}_{\cdot j} + SE_{\cdot j}]$. **c** A durability test in which the apparent (or average) contact load is held constant ($\bar{F} \approx 70F_{off,0}$, point "o" in **a**), where the drastically enhanced adhesion strength is maintained for over $2 \times 10^7$ vibration cycles without any noticeable degradation in the adhesion performance.

modulation experiments do not require a preload, which we attribute to the instantaneously formed reliable adhesion by the assistance of micro-vibration.

**Continuous regulating capability.** To experimentally show the capability of continuously tuning surface adhesion (both enhancing and weakening) by using our method, we varied the frequency of micro-vibration ($\omega/(2\pi)$) from 0.1 to 0.8 kHz and

changed the amplitude ($A_b$) from 5 to 110 μm. The experimental data are shown by the plus sign in Fig. 3. For comparison, the theoretical predicted pull-off forces (based on Eq. (3)) $F_{off}/F_{off,0}$ are plotted as contours, where the red contour corresponds to the maximum of $F_{off}/F_{off,0}$. It is clear that the trend of the experimental data agrees with the theoretical results, especially for large frequencies, while a deviation occurs when $\omega/(2\pi) < 400$ Hz. This is because Eq. (3) only holds for $\omega \gg 0$ (see the Discussion section). However, the lower and upper bounds can be theoretically determined as 0 and 78, respectively (see the Discussion section), which agree well with the experimental results, where $\min\{F_{off}/F_{off,0}\} = 0$ and $\max\{F_{off}/F_{off,0}\} = 76.83$. The observed continuous tuning of surface adhesion, from weakening ($F_{off}/F_{off,0} < 1$) to enhancement ($F_{off}/F_{off,0} > 1$), is remarkable since in many practical applications, such as those using gecko-like robots, interface adhesion, and debonding, must be executed repeatedly.

**Adhesion switching and energy consumption.** From the viewpoint of practical application, the timescale of adhesion switching is another key parameter and can be measured by a sudden change in the actuation amplitude, frequency, or apparent contact load. Typical results are shown in Supplementary Note 5, where we measured the adhesion by a sudden change either in the actuation amplitude (e.g., from $A_b = 30$–80 μm) or in the apparent contact load (e.g., from $\bar{F}/F_{off,0} = 6$–69) with a constant actuation frequency of 450 Hz. We observed that the system responds very quickly (within 16 ms, see Supplementary Fig. 2 and Supplementary Table 1). Similarly, by an abrupt increase in the actuation frequency from $\omega/(2\pi) = 300$–500 Hz, but with a constant actuation amplitude of 30 μm, the measured shortest transformation time is 33 ms (Supplementary Table 1). All of the measured adhesion switching times are comparable to that of geckos (15 ms[3]), which indicates that our method is desirable for applications, such as fast mobile robotics and space technologies that require the ability to move and place objects accurately and quickly. Considering that the input power for adhesion regulation is crucial for practical applications, we further calculated the input power. By varying the actuation amplitude from 8.6 to

96.5 μm with a constant actuation frequency of 450 Hz, we found that the input power per contact area of the system varies from 0.12 to 5.57 mW mm$^{-2}$ (see Supplementary Note 6 and Supplementary Table 2), which is very low and promising for highly integrated and untethered applications.

## Discussion

To understand the observed phenomena, we modeled the adhesion system as sketched in Fig. 1b. Mechanically, the stiff contactor/vibrating soft platform system (left) is equivalent to a vibrating soft contactor/stiff platform system (right; see Supplementary Fig. 3 and Supplementary Movies 1 and 2). Under an external force, $F$ (pressing is defined as positive and pulling as negative), the stiff part will penetrate into the soft part. The penetration depth is defined as $\delta$. In our experiments, a spherical crown-shaped stiff glass (with a curvature radius of $R$) and a stiff connector (the part between the rubber band and the glass in Fig. 1a) served as the contactor (with a mass of $M$), and a widely used soft material, PDMS, was chosen as the soft platform. Poisson's ratio of PDMS can be taken as $\nu \approx 0.5$. The elastic modulus of PDMS is thus $K = 4/3E/(1 - \nu^2) \approx 16E/9$[51], where $E$ is Young's modulus. Considering that the ratio of the vibration amplitude of the contactor to the elongation of the rubber band in our experiments was <0.01, we assumed that the pulling force in the rubber band measured by the force sensor approximately equals the apparent/average contact load $\bar{F}$. We herein derived the governing equations for the dynamic adhesive contact as (see Supplementary Note 1):

$$\begin{cases} \frac{Ka}{2}\left(3\delta - \frac{a^2}{R}\right) + M\ddot{\delta} + c\dot{\delta} = F = \bar{F} + f, \\ \frac{3K}{8\pi a}\left(\delta - \frac{a^2}{R}\right)^2 = w(a, \nu), \end{cases} \quad (1)$$

where $f = A_f \cos \omega t$ is the imposed micro-oscillation of $\bar{F}$. In our experiments, $A_f = MA_b\omega^2$, where $A_b$ and $\omega$ are the amplitude and circular frequency exerted on the platform by the loudspeaker, respectively. $c$ is the damping coefficient of the system. $w(a, \nu)$ is the effective adhesive work at the contact line, which asymmetrically depends on the speed of the moving crack tip $\nu$ ($\nu \approx -\dot{a}$). Specifically, as the crack propagation speed ($\nu$) increases, the adhesion work first increases exponentially, then falls sharply and finally rises rapidly again. As the healing speed ($|\nu|$) increases, the adhesion work rapidly approaches zero (Fig. 4). When $\nu \leq \nu_c$, $w(a, \nu)$ is (see Supplementary Note 1)[24–30,52–61]:

$$w(a, \nu) = \begin{cases} w_0\left(1 + \left|\frac{\nu}{\nu_0}\right|^\alpha\right), & 0 \leq \nu \leq \nu_c, \\ w_0\left(1 + \left|\frac{\nu}{\nu_0}\right|^\alpha\right)^{-1}, & -\infty < \nu \leq 0, \end{cases} \quad (2)$$

where $w_0$ is the quasi-static (intrinsic) adhesion work of the contact pair, $\nu_0$ and $\alpha$ are two constants that depend on the materials in contact, and $\nu_c$ is the saturation velocity; $w$ locally decreases with $\nu$ when $\nu > \nu_c$ (Fig. 4). If ignoring the time-dependent terms, Eq. (1) degenerates to the classic JKR (Johnson–Kendall–Roberts) theory[51,52], that is, $\delta = a^2/R - \sqrt{8/3 \cdot \pi a w/K}$ and $F = Ka/2 \cdot (3\delta - a^2/R)$. To theoretically predict the effect of micro-vibration on the contact adhesion, we approximately solved Eq. (1) (see Supplementary Note 2). The obtained apparent contact load as function of the average contact radius ($\bar{a}$) and the micro-vibration ($A_f$, $\omega$) reads

$$\bar{F}(A_f, \omega, \bar{a}) = \frac{K\bar{a}^3}{R} - \sqrt{6\pi w_0 K\bar{a}^3}$$
$$- \frac{3}{2}\frac{A_f K\bar{a}}{\sqrt{(3K\bar{a}/2 - M\omega^2)^2 + c^2\omega^2}}; \quad A_f \geq 0, \omega \gg 0, \nu \leq \nu_c. \quad (3)$$

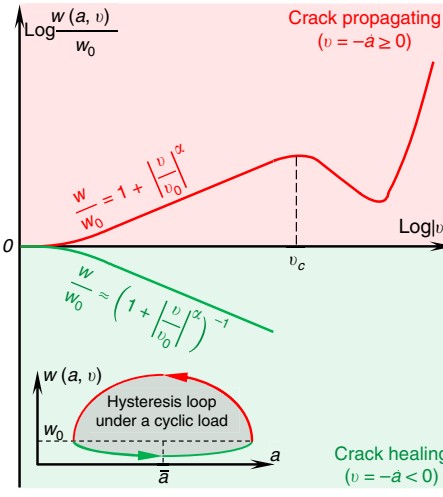

**Fig. 4 The constitutive relation of effective adhesive work.** The propagating and healing of an interface crack generally have different resistances, which results in the adhesion hysteresis. The inset shows a typical hysteresis loop under a cyclic load. For the situation of $\nu \leq \nu_c$, where $\nu_c$ is the saturation velocity, the adhesion hysteresis can be utilized to increase the apparent adhesion work, hence enhance the interfacial adhesion strength. The experimentally determined $\alpha$ varies between 0.1 and 0.8, and the "N" shaped $w(\nu)$ curve is generally observed when $\nu > \nu_c$[25–27,30]. More details can be found in Supplementary Note 1.

Based on Eq. (3), the pull-off force ($F_{off}$) can be readily obtained by finding the minimum value of $\bar{F}$ for $\nu \leq \nu_c$. Equation (3) also indicates that $F_{off}$ is higher than $F_{off,0} = -3/2 \cdot \pi R w_0$ ($F_{off} = F_{off,0}$ when $A_f = 0$) if $A_f > 0$ (under micro-vibration), which we attributed to the adhesion hysteresis (see the hysteresis loop in Fig. 4) induced by the adhesion asymmetry as stated above. In fact, a similar "vibration suction method"[62–64] has been reported to enhance the negative pressure through vibration.

On the other hand, Eq. (3) allows us to quantify the influence of micro-vibration on the apparent contact load. Typical results are shown in Fig. 5, where the apparent contact load $\bar{F}$ is normalized by $F_{off,0}$. It is clear that even for a very small increase in $A_b$ ($A_f = MA_b\omega^2$, with fixed $2\pi/\omega = 450$ Hz), both the ranges of the apparent contact load and the contact radius increase significantly, which reveals that the surface adhesion can be effectively regulated by the micro-vibration. In principle, for any given combinations of $A_b$ (or $A_f$) and $\omega$, the maximum value of $\bar{F}/F_{off,0}$ gives the normalized apparent pull-off force $F_{off}/F_{off,0}$ (Supplementary Note 3). As the normalized pulling force increases, the average contact radius decreases/increases along the $\bar{F}/F_{off,0} \sim \bar{a}$ curve until it reaches the maximum normalized pulling force (blue circles in Fig. 5a), at which interface adhesion fails and the two contact parts are pulled off. Therefore, the upper bound of $F_{off}/F_{off,0}$ can be obtained by setting $\max\{\nu\} = \nu_c$, which gives $F_{off}/F_{off,0} = |\nu_c/\nu_0|^\alpha/4$ (Supplementary Note 3, red dashed line in Fig. 5a). The lower bound of $F_{off}/F_{off,0}$ is 1, which is set by letting $A_f = 0$. In practice, we can regulate the contact adhesion by keeping either $A_b$ or $\omega$ fixed while varying the other to obtain a target pull-off force. Typical results are shown in Fig. 5b, where the blue curve in region 1 is obtained by fixing $\omega/(2\pi) = 450$ Hz. We found that the pull-off force in this region depends almost linearly on the micro-amplitude, that is, $F_{off}/F_{off,0} \propto A_b$, in line with the theoretical prediction (Eq. (3) and Supplementary Note 3).

Notably, the results above are based on the assumption that the crack propagation speed ($\nu \approx -\dot{a}$) is lower than the saturation velocity $\nu_c$. In some cases, if the combinations of $A_b$ and $\omega$ result

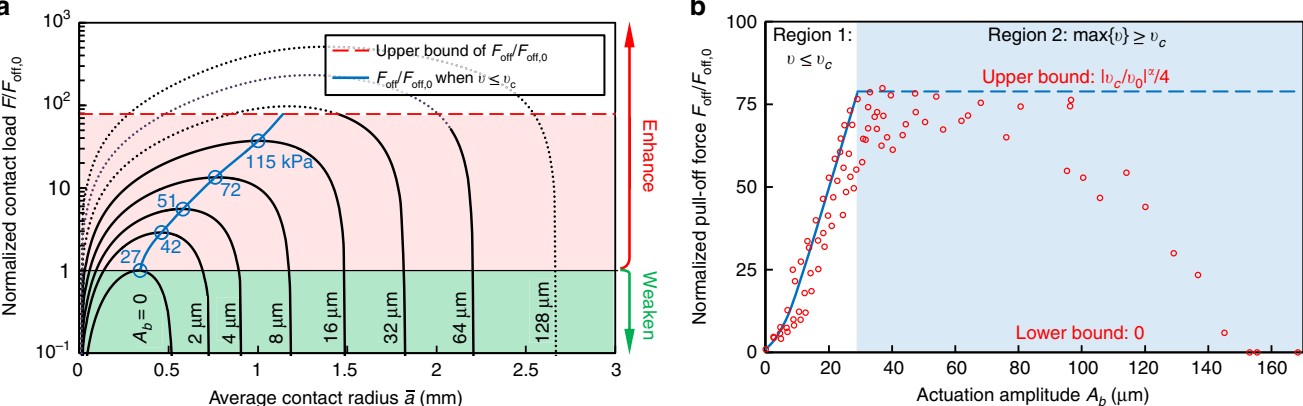

**Fig. 5 Typical results for the regulation of contact adhesion by micro-vibration with a frequency of 450 Hz. a** Dependence of $\bar{F}/F_{off,0}$ on the vibration amplitude $A_b$ ($A_f = MA_b\omega^2$) and the average contact radius $\bar{a}$. The dotted lines correspond to max{$v$}>$v_c$ and the solid lines correspond to $v \leq v_c$. The red dashed line is the upper bound of $F_{off}/F_{off,0}$ (i.e., $|v_c/v_0|^\alpha/4$, Supplementary Note 3). The blue curve gives the dependence of $F_{off}/F_{off,0}$ on $\bar{a}$ by connecting the maximums of all the $\bar{F}/F_{off,0} \sim \bar{a}$ curves (blue circles, $v \leq v_c$). The numbers (unit: kPa) near the blue circles are the corresponding theoretical adhesion strengths calculated by $\bar{F}/\pi\bar{a}^2$, which can be greater than that of a gecko (100 kPa[1,8]). **b** Dependence of $F_{off}/F_{off,0}$ on the actuation amplitude $A_b$. The blue curve in Region 1 is obtained based on Eq. (3), where $v \leq v_c$. Once max {$v$} > $v_c$, $F_{off}/F_{off,0}$ can be varied within a wide range between the lower bound and the upper bound (Region 2). The "o" marks indicate experimental results, which agree with the predicted blue curve in region 1 or are exactly located in the predicted range in region 2. The parameters for the theoretical calculations in **a**, **b** are based on our experiments. The thickness of the PDMS platform is 3 mm, and the corresponding effective elastic modulus is higher than the actual elastic modulus and determined as $K \approx 6.211 \times 2.23 = 13.85$ MPa based on finite element simulation (see Supplementary Note 4 and Supplementary Fig. 1; the actual elastic modulus of the PDMS is $K = 16E/9 = 6.211$ MPa corresponding to an infinite thickness). The radius and mass of the contactor are $R = 51.64$ mm and $M = 2.905$ g, respectively. The damping coefficient $c \approx 5.1$ N s m$^{-1}$ is determined by the power consumption data. $w_0 = 40$ mJ m$^{-2}$, $v_0 = 0.6$ μm s$^{-1}$, $\alpha = 0.46$, and $v_c = 0.16$ m s$^{-1}$ are experimentally measured by the peeling test.

in max{$v$} > $v_c$, the adhesion system will become dynamically instable and the contact states cannot be predicted by Eq. (3) (shown as dots instead of solid lines in Fig. 5a). Although it is difficult to precisely predict the contact states when max{$v$} > $v_c$, the upper bound of $F_{off}/F_{off,0}$ should be the same as $|v_c/v_0|^\alpha/4$ (red dashed line in region 2, Fig. 5b) since the rate-dependent effective adhesion work decreases when $v$ is larger than $v_c$ (Fig. 4). This relationship has been verified by experiments in which the actuation frequency was kept constant while the amplitude was increased from 2.4 to 168 μm (denoted by "o" in Fig. 5b). Notably, the adhesion strength can be reduced to 0% of the default/quasi-static/intrinsic strength when $A_b$ >153 μm (Figs. 3 and 5b). The dynamic instability (see the "N"-shaped constitutive curve in Fig. 4) induced adhesion deterioration demonstrates that our method can not only enhance the surface adhesion (pink region in Fig. 5a) but also reduce the contact adhesion to make the originally sticky interface a non-stick interface by increasing the vibration frequency or amplitude so that $v \gg v_c$ (green region in Fig. 5a). This is very attractive for applications such as the using of robots for the nondestructive separation of adhesion interfaces. Experimentally, we found that to make $F_{off}/F_{off,0}$ <1, one needs to set a sufficiently high $A_f$ or $\omega$ to effectively weaken the adhesion strength (Figs. 3 and 5b).

## Methods

**Adhesion strength regulation test.** The experiments were carried out with a custom-built adhesion testing system (Fig. 1), where a stiff glass contactor with a spherical surface was aligned perpendicular to a flat PDMS platform. The PDMS platform was prepared by pouring mixed liquids (Dow Corning SYLGARD® 184, with a base to cross-linker weight ratio of 10:1) into a shallow rectangular slot mold (50 × 20 ×3 mm³) at room temperature, and the upper surface became very smooth under the action of gravity and surface tension. After curing at 23 °C for 720 h, the final thickness of the PDMS platform was measured to be 3 mm. Two electro-dynamic coil cone loudspeakers (with a diameter of 8 in., rated power of 200 W, and impedance of 8 Ω) were used to excite the vibration of the PDMS platform. The loudspeakers were driven by a harmonic signal generator (with working frequency <20 kHz), equipped with a power amplifier (with working frequency <10 kHz and power of 1 kW). The vibration amplitude was controlled by adjusting the

amplification factor of the power amplifier. An optical plano-convex lens (with a radius of curvature of 51.64 mm) was chosen as the spherical crown-shaped glass contactor, which was then connected in series with a rubber band buffer (cut from a yellow natural latex loop; section of 1.5 × 1.5 mm², stiffness of 121.4 N m$^{-1}$) and a force sensor. The force sensor (with a measurement range of ±2 N, resolution of 30 μN, and working frequency <2 kHz) was constructed by a cantilever beam with several strain gauges adhered to the surface and was used to measure the normal force near the free end. The force sensor was then fixed to the slider of a vertical translation stage. During the adhesion test, the stiff contactor was precisely controlled to move along the normal direction of the PDMS platform by actuating the vertical translation stage. The loading rate was varied between 3 μm/s and 0 with holding times of 1 s and 10 s, respectively (such a loading mode approximates the quasi-static). The displacements of the platform and the contactor were measured by four laser displacement sensors (KEYENCE® LK-G5001, LK-H150) with a sampling frequency of 100 kHz, which were recorded by a data acquisition card (NI USB-6212). The size of the adhesion region was recorded by a high-speed camera at a frame rate of 3500 frames/s. During the adhesion test, the temperature and humidity were monitored and maintained at 23 °C and 65% relative humidity, respectively.

**Durability test.** As the vibration continued, the contact load was slowly increased by moving the cantilever beam upwards (the loading rate was set as 3 μm s$^{-1}$) with the vertical translation stage to a target value, and then holding the cantilever beam position with the translation stage by simply turning its power off. During the tests, both the displacement and the force were recorded until the contactor separated from the PDMS platform spontaneously.

**Determination of material properties.** The Young's modulus of the PDMS ($E$) was measured by dynamic thermomechanical analysis and compared with the corresponding value reported in the literature[65]. The typical value is taken by averaging the modulus at ~450 Hz. The effective elastic modulus was determined to be 2.23 times the actual elastic modulus (see Supplementary Note 4 and Supplementary Fig. 1). The stiffness of the rubber band was measured by a uniaxial tension test. The damping coefficient of the system ($c$) was determined by the power consumption data with $v < v_c$, and the typical value was determined by averaging the coefficients at ~450 Hz. Specifically, based on the recorded data, we calculated the experimental power consumption by $P = \omega/(2\pi) \cdot \oint f \mathrm{d}\delta$. By comparison to the relation $P = c\omega^2/2 \cdot A_f^2 / \left[ (3K\bar{a}/2 - M\omega^2)^2 + c^2\omega^2 \right]$ (see the main text), $c$ was thus obtained. The adhesion properties ($w_0$, $v_0$, and $v_c$) were measured by a peeling test as follows: Preparing bilayered PDMS tape. A rectangular slot mold (600 × 50 × 0.25 or 0.5 or 1.0 mm³) was first sealed by a commercial transparent tape (with width of 60 mm), and then the as-mixed PDMS liquids were

poured into the mold. After solidification and demoulding, the PDMS and commercial transparent tape were glued together, resulting in a bilayered PDMS tape. The prepared bilayered PDMS tape was then cut into four narrower tapes with a size of $600 \times 10 \times 0.25/0.5/1.0$ mm$^3$ and then adhered to a cleaned glass plate. Slight pressure was applied to ensure good adhesion between the PDMS tape and the glass plate. Placing the glass plate vertically and then peeling off the upper end of the PDMS tape slightly, this end of the PDMS tape was then connected to a free weight by a stiff string. Once the weight was released, the weight fell vertically under gravity, and the PDMS tape peeled off from the glass surface with a peeling angle of 180°. By recording the falling displacement and time, the falling speed of the weight can be determined. By substituting these values into the Kendall formula (the peeling force is approximately $wb/2$ at the peeling angle of 180°, where $w$ is the work of adhesion and $b = 10$ mm is the width of the PDMS tape), and applying the constitutive relation of $w = w_0(1 + |v/v_0|^\alpha)$ (Eq. (2)), we obtained the adhesion property constants by least square fitting ($w_0$ was measured by the adhesion strength regulation test with a vibration amplitude of zero, i.e., without vibration).

## Data availability

Data supporting the findings of this manuscript are available from the corresponding authors upon reasonable request.

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

## Acknowledgements

We thank Yongshou Liu (Northwestern Polytechnical University, People's Republic of China) for providing the laser displacement measurement system. We also thank Liangliang Zhu (Northwest University, People's Republic of China), Xiangbiao Liao (Columbia University, USA), and Chao Lu (Columbia University, USA) for scientific discussions. We acknowledge supports from the Fundamental Research Funds for the Central Universities (413000094), the National Key Research and Development Program of China (2016YFB0700300), the National Natural Science Foundation of China (Nos. 11602175, 11632009, 11672247, 11872302, and 11902226), and the Key Research and Development Program of Shaanxi (2018ZDXM-GY-131).

## Author contributions

L.S. and X.C. conceived the idea. X.C. supervised the project. L.S., L.J., and Z.L. designed the experiments. L.J., H.L., J.G., and Z.G. carried out the experiments. L.S. and Z.L. developed the theoretical model. L.S., L.J., Y.L., Z.L., and X.C. analyzed the data and wrote the manuscript.

## Competing interests

The authors declare no competing interests.
