## [Peer Review File · Nature Communications]

Reviewers' comments:

Reviewer #1 (Remarks to the Author):

This paper describes a method for modulating the adhesion between a suspended glass indenter and an adhesive substrate via vibration of the substrate. Depending on the frequency and amplitude of vibration, adhesion can be increased substantially above the static case ($\sim 70\times$) and slightly below ($0.6\times$). A model describing this behavior is presented and data is shown that supports the model. The time for switching between high and low adhesion is tested and shown to be around 15-30ms. The authors claim potential applications in gecko-inspired climbing robots.

While this work presents a very intriguing concept (vibration-modulated adhesion), this reviewer has major concerns that would need to be addressed before publication.

First, it is unclear whether the applicability of the concept is as broad as proposed. The system that is tested and modeled has a spherical indenter suspended by a rubber band, through which load is applied, while the adhesive substrate is vibrated. Translating this to a practical system might not be trivial. For instance, for a climbing robot, would the vibration element be contained in the foot of the robot? Is that practical? Will the flat-flat contact that would be expected in a climbing robot behave the same? Would the high shear loads (the primary load for a climbing robot) affect the results? To this reviewer, either these questions need to be addressed or the emphasis on this application should be removed. Ideally, the authors would show a simple demonstration of an adhesive pad suspending different masses depending on vibration characteristics. Further, only a small change in adhesion is seen between the default adhesion and a reduced adhesion state, meaning to make a "switchable" adhesive, the vibration would have to be continuously on for the "strong" state. This could be a limitation for robots that are designed to remain passively on the wall for long periods of time.

Second, the purported impact on climbing robots and gecko-inspired adhesives is likely overstated, given the lack of appropriate references. It is claimed "Adhesion switching has been proposed by either altering the load path and failure mode of the interface to form a "smart/phase change" interface or controlling the debonding speed, yet only two adhesion states, i.e. "strong" or "weak", can be identified and the switching speed is intrinsically limited." This misses a whole class of adhesives, known as "controllable," inspired by the gecko [for instance: R1, R2, R3, R4]. The normal adhesive force can be continuously tuned by the direction of the applied load. Further, these adhesives have been shown to switch on the order 20ms, roughly the same rate as reported in the presented work [R5].

Third, the work is not presented clearly. The most important part of the work is a simple and clear explanation of why this effect occurs. However, besides presentation of data showing the asymmetry of growth and retraction of the contact patch, a clear explanation is not provided. In a single sentence, why does this work? More generally, the organization and presentation are not clear. This is in addition to consistent English errors throughout.

Overall, while this work is intriguing, it is not ready yet for publication in a journal of the caliber of Nature Communications. A much tighter, more clear, well crafted manuscript is needed that carefully considers the proposed applications and explains simply the working concept.

References:

R1: Autumn K, Dittmore A, Santos D, Spenko M, Cutkosky M. Frictional adhesion: a new angle on gecko attachment. *Journal of Experimental Biology*. 2006 Sep 15; 209(18):3569-79.

R2: Parness A, Soto D, Esparza N, Gravish N, Wilkinson M, Autumn K, Cutkosky M. A microfabricated wedge-shaped adhesive array displaying gecko-like dynamic adhesion, directionality and long lifetime. *Journal of the Royal Society Interface*. 2009 Mar 18; 6(41): 1223-32.

R3: Ruffatto III D, Parness A, Spenko M. Improving controllable adhesion on both rough and smooth surfaces with a hybrid electrostatic/gecko-like adhesive. *Journal of The Royal Society Interface*. 2014 Apr 6; 11(93): 20131089.

R4: Northen MT, Greiner C, Arzt E, Turner KL. A Gecko-inspired reversible adhesive. *Advanced Materials*. 2008 Oct 17; 20(20): 3905-9.

R5: Hawkes EW, Jiang H, Cutkosky MR. Three-dimensional dynamic surface grasping with dry adhesion. *The International Journal of Robotics Research*. 2016 Jul; 35(8): 943-58.

Reviewer #2 (Remarks to the Author):

What are the major claims of the paper?

The major claims of the paper are that the authors have developed a (1) robust and (2) predictable method of micro vibration-based adhesion. The authors develop a model to predict adhesion performance, to (3) continuously change the strength of adhesion. Furthermore, the authors state that this is a (4) simple and practical way to modulate adhesive strength.

Findings:

The authors find that the adhesion strength can be varied by two orders of magnitude (100x) using either vibration frequency or amplitude.

For a specific range of loading and actuation conditions, they find that the timescale of switching is comparable to that found in biological systems such as the gecko.

For a quasi-static loading case, they demonstrated that their adhesion system was able to hold a load 70x greater than the non-vibrating case. They held this loading case for 2×10^7 vibration cycles or approximately 12 hours without any degradation in performance.

They calculate input power for the system theoretically and find that the power/area is less than 1 mW/mm².

Are they novel and will they be of interest to others in the community and the wider field?

These findings appear to be novel and I believe that they would be of interest to the community and beyond. A significant amount of work is put toward the development and presentation of a model of the phenomena.

Is the work convincing, and if not, what further evidence would be required to strengthen the conclusions?

The work appears to be convincing, but there are some gaps in the presentation of the results and the

subsequent discussion. Prior to publication, shortcomings mentioned in the major and minor issues sections should be addressed.

We would also be grateful if you could comment on the appropriateness and validity of any statistical analysis, as well as the ability of a researcher to reproduce the work, given the level of detail provided. It is unclear how repeatable these results are between tests. Only one graph shows error bars in the whole paper, and the data for those error bars was taken from one test.

Figure 1 does a good job showing the experimental setup that is used to generate the results that are presented. However, more information on the brand or type of force sensor used, as well as the stiffness/material of the cantilever beam would be helpful to someone trying to recreate the study. The methods section should be updated to include parameters about the loudspeakers such as their size, impedance, resistance, and how the amplitudes of vibration were synced.

Major Issues

1. The study fails to address how the findings relate to previous research in this area. The authors should rewrite their introduction and discussion to reference the most closely related literature on adhesion for robotics, and clearly explain the relative advantages and disadvantages of the proposed approach.
2. The authors never directly show the forces they can achieve, but only show a relative increase or decrease in performance. However, it is important to present the base-line pull-off data, which is never explicitly presented or calculated. For a discussion of practicality, which is a main claim of this paper, it is important to discuss whether or not this system generates enough adhesion to be useful.
3. Fig 2b: The wording used to describe this test is confusing, specifically pull-off. I assume this test was performed by using the quasi-static loading methodology discussed later in the paper to achieve a 70x improvement and then it was left to vibrate. If this is the case, full pull-off never occurs during this test. The only pull-off actually performed is for the non-vibrating case.
4. Fig 3b: It is difficult to assess if the model is a good match as region one only possesses two data points. Please present some measure of the variation for each experimental trial and/or more data points.
5. Fig 4: There are relatively few experimental samples shown for high frequency and high amplitude cases. Data should be presented for these cases or a rationale provided for why these tests were not performed. What was the sampling strategy for this set of experiments?
6. Fig S3: It is not always obvious when the test starts and when the test ends from inspecting the graphs. Please improve the clarity.
7. In the supplemental materials, section I, the equation S1 is presented as being based on previous work, but the authors list 15 different citations in support of this claim. Is it true that all 15 papers directly address this claim? Perhaps the authors could be a bit more precise in their citations to guide the reader in their review of this previous work.
8. Lines 7-8: The abstract claims two orders of magnitude improvement, but the reported data does not appear to support this conclusion.
9. Line 125: How was this verified? What observation did the authors make?
10. Lines 150-151: The load is increased from 35x to 45x. What would it look like for a more dramatic change such as going from 7x to 70x?

Minor Issues

1. The authors should generally revise the language of the manuscript to improve readability and clarity.
2. Fig 3a: Can the failure points indicated by the blue line be verified using experiments?
3. Line 64: Please cite or explain the source of this relation.
4. Line 148: Reference to fig S4, but should be S3
5. Table S2: Is the power presented in this table a theoretically or experimentally determined value?

Please specify.

Reviewer #3 (Remarks to the Author):

Review: "Rapid and continuous regulating" Shui et al.

The authors introduce a method by which interfacial adhesion can be regulated by exciting mechanical "micro-vibration" perpendicular to the contact plane. They report that apparent adhesion strength can be varied by two orders of magnitude by control of frequency and amplitude of vibrations. Switching can be rapid (~ 15 ms) and repeated more than 10^7 times. The method requires no special microstructure. This could be a very important paper but several things about how the experiments were done are unclear to me and parts of the theoretical model need to be better explained. In addition, the outline of the theory needs some more physical explanation to explain the effect. So, I believe that the manuscript needs a significant revision before being reviewed again. To repeat, this is potentially an important and novel paper but, as written, I am unable to decide if that is the case. I would ask the authors to address the following points.

- 1) Language needs some work.
- 2) Some of the words in Figure 1 are very difficult to read.
- 3) I don't follow the caption of Figure 2b. It purports to show pull-off force for 2×10^7 cycles but at the same time it is stated that contact radius barely changed (Fig. 2a). I would like to see a full force-displacement curve to go with the Fig. 2a. Is it that the normal force went up transiently to 70 times the pull-off force without vibration? This is critical point to explain about how pull-off forces were measured.
- 4) Similarly in the Methods section (near line 175), exactly how the pull-off force was measured is not at all clear. Again, it is important to show the measured force-displacement data so that we can witness the reported increase in pull-off force from this measurement.
- 5) On line 54, the asymmetry of radius change could occur due to other reasons such as effect of rate on adhesion energy, bulk viscoelasticity, inertia, etc. How do the authors ascribe it to the difference between crack growth versus closure?
- 6) Line 64, to which elastic modulus are the authors referring (K)?
- 7) Equations (2) and (S1) are not the same. In SI, equation (S1) is introduced by referring to many papers. However, since this is a critical part of the paper, the physical basis for this interfacial constitutive law must be explained, at least in SI. For example, the model assumes that at very slow velocities the work of opening and closing the interface is the same, but this is usually not true for Sylgard 184 PDMS even at slow enough rates that the works of adhesion show no rate effect.
- 8) Please provide a reference for the Lagrange equations of motion for nonconservative systems. Please provide references to the form of elastic strain energy given on page 2/12 of SI. The specification in eq. (S1) that the first two equations apply only at $r=a$ is not clear to me and so it is also not clear because of this definition how the derivatives with respect to a in equation (S2) have been carried out. This needs to be clarified. It is not clear how the red line in Figure S1 relates to the first part of eq. (S1), which does not tell us what happens for positive velocities greater than v_c .
- 9) It seems that a and δ being treated as independent kinematic quantities? In contact mechanics they are usually related.
- 10) The claim in line 160 that the authors have demonstrated a method for modulating adhesion that works for "any adhesive materials (sic)" is too broad. Their demonstration depends on a particular rate dependence of adhesion and the effect has been shown only for one materials system.
- 11) I do not follow how the authors get S4 from S3. There is some approximation involved that has not been stated, it seems. Please clarify.

- 12) The authors quote but don't discuss the paper by Wahl et al. (ref. 23) which examined a closely related experiment in which I don't believe any significant enhancement of pull-off force was reported. The authors need to compare their work with that of ref 23.
- 13) For these reasons, I have been unable to check all the equations in SI.
- 14) Some more detailed and better physical explanation for this effect is needed.

Responses to Reviewer #1

This paper describes a method for modulating the adhesion between a suspended glass indenter an adhesive substrate via vibration of the substrate. Depending on the frequency and amplitude of vibration, adhesion can be increased substantially above the static case (~70x) and slightly below (0.6x). A model describing this behavior is presented and data is shown that supports the model. The time for switching between high and low adhesion is tested and shown to be around 15-30ms. The authors claim potential applications in gecko-inspired climbing robots.

While this work presents a very intriguing concept (vibration-modulated adhesion), this reviewer has major concerns that would need to be addressed before publication.

Response: We thank you very much for thinking our work very intriguing.

First, it is unclear whether the applicability of the concept is as broad as proposed. The system that is tested and modeled has a spherical indenter suspended by a rubber band, through which load is applied, while the adhesive substrate is vibrated. Translating this to a practical system might not be trivial. For instance, for a climbing robot, would the vibration element be contained in the foot of the robot? Is that practical? Will the flat-flat contact that would be expected in a climbing robot behave the same? Would the high shear loads (the primary load for a climbing robot) affect the results? To this reviewer, either these questions need to be addressed or the emphasis on this application should be removed. Ideally, the authors would show a simple demonstration of an adhesive pad suspending different masses depending on vibration characteristics. Further, only a small change in adhesion is seen between the default adhesion and a reduced adhesion state, meaning to make a “switchable” adhesive, the vibration would have to be continuously on for the “strong” state. This could be a limitation for robots that are designed to remain passively on the wall for long periods of time.

Response: We thank you very much for these constructive suggestions. They are

great helpful to improve the value and quality of the work. To address your questions, we have added a simple demo (refer to **Video 1** in the supplementary materials) as suggested. In the demo: (a) The vibration elements (here are small converted electromagnetic loudspeakers) are contained in the contactors. In such scheme, the vibration element is contained in the “foot” of the “robot”. (b) It is almost a flat-flat contact (the contactor has in-plane diameter of 15 mm and out-of-plane curvature radius of 80 mm, i.e. height difference between edge and center of the contactor is about 0.35 mm). Specifically, three contactors covered by commercial thin PVC anti-slip mats are fixed to a triangular plastic frame. When turn on the vibration elements, the plastic frame with the contactors adheres to a flat glass wall well, while once turn off the vibration elements, the plastic frame falls by gravity (see **Video 1** for details). To study the effects of shear loads on the adhesion, we simply rotate the adhesion system in **Video 1** by 90 degrees, which mimics the climbing robots, we found that the micro-vibration can still modulate the interface adhesion well though shear loads (due to the gravity of the “robot” in **Video 2**) exerted on the interface.

As for the comments of “only a small change in adhesion is seen between the default and a reduced adhesion state”, we have added new experiments to show significant changes in adhesion (see revised Figs. 3b and 4), the adhesion strength can be reduced to 0% of the default/quasi-static/intrinsic one.

As the limitation of “the vibration would have to be continuously on for the ‘strong’ state” commented by the referee, we admit that, but fortunately, we found that the system has very low power consumption ($\sim 10^0$ mW/mm², refer to the 2nd to last paragraph of the main text), and thus it is expected to be competent for most of applications.

Second, the purported impact on climbing robots and gecko-inspired adhesives is likely overstated, given the lack of appropriate references. It is claimed “Adhesion switching has been proposed by either altering the load path and failure mode of the interface to form a “smart/phase change” interface or controlling the debonding speed, yet only two adhesion states, i.e. “strong” or “weak”, can be identified and the switching speed is intrinsically limited.” This misses a whole class of adhesives, known as “controllable,” inspired by the gecko [for instance: R1, R2, R3, R4]. The normal adhesive force can be continuously tuned by the direction of the applied load.

Further, these adhesives have been shown to switch on the order 20ms, roughly the same rate as reported in the presented work [R5].

Response: Thank you for suggesting these important references. We have added them to our References. We completely agree with the actual characteristics of the class of adhesives that “the normal adhesive force can be continuously tuned by the direction of the applied load”. The suggested references have been added to the manuscript as ref. 5, 13-16. In addition, the original description about “Adhesion switching” has been revised as (refer to 2nd paragraph of the main text): “Considering that the apparent adhesion strength can be affected by the real contact area, the size of contact region ¹⁷, and the adhesion work ²⁰, adhesion switching has been proposed by either controlling the load path and failure mode of the interface or constructing a “smart” interface such as phase transition interface ^{6,20}, based on which diversity of adhesion switching triggers e.g. mechanical ^{12, 15, 21-30}, electro/magnetic ^{14, 16, 31}, lighten ³²⁻³⁴, fluidic ³⁵⁻³⁸, thermal ³⁹⁻⁴⁴, have been reported. Note that the gecko-inspired directional adhesion ^{12, 13, 15, 16, 45-50} and the debonding/peeling speed-regulated adhesion ²⁴⁻³⁰ provide the promising methods to tune the adhesion strength continuously and rapidly. However, at present, there is still lacking of simple and long-term effective adhesion modulation method. In fact, in most cases, only two adhesion states, i.e. “strong” or “weak”, can be identified, or easy to de-adhere but hard to re-adhere rapidly, or the switching speed is intrinsically limited ^{6,20}.”

Third, the work is not presented clearly. The most important part of the work is a simple and clear explanation of why this effect occurs. However, besides presentation of data showing the asymmetry of growth and retraction of the contact patch, a clear explanation is not provided. In a single sentence, why does this work? More generally, the organization and presentation are not clear. This is in addition to consistent English errors throughout.

Response: We have revised our manuscript to make the presentation clear and tried our best to correct the English errors. In a single sentence, the proposed method works because the interfacial adhesion work significantly and asymmetrically depends on the speed of the moving crack tip (the edge of the contact interface), based on which we introduce a micro-vibration to the adhesion system to effectively modulate the moving speed of crack tip.

Overall, while this work is intriguing, it is not ready yet for publication in a journal of the caliber of Nature Communications. A much tighter, more clear, well crafted manuscript is needed that carefully considers the proposed applications and explains simply the working concept.

Response: We have rewritten the introduction and the discussion parts. Considerable revisions have been made throughout the manuscript to clarify the working concept. We believe the revised version has been significantly improved based on the referee's comments/suggestions.

Responses to Reviewer #2

What are the major claims of the paper?

The major claims of the paper are that the authors have developed a (1) robust and (2) predictable method of micro vibration-based adhesion. The authors develop a model to predict adhesion performance, to (3) continuously change the strength of adhesion. Furthermore, the authors state that this is a (4) simple and practical way to modulate adhesive strength.

Findings:

The authors find that the adhesion strength can be varied by two orders of magnitude (100x) using either vibration frequency or amplitude.

For a specific range of loading and actuation conditions, they find that the timescale of switching is comparable to that found in biological systems such as the gecko.

For a quasi-static loading case, they demonstrated that their adhesion system was able to hold a load 70x greater than the non-vibrating case. They held this loading case for 2×10^7 vibration cycles or approximately 12 hours without any degradation in performance.

They calculate input power for the system theoretically and find that the power/area is less than 1 mW/mm^2 .

Are they novel and will they be of interest to others in the community and the wider field?

These findings appear to be novel and I believe that they would be of interest to the community and beyond. A significant amount of work is put toward the development and presentation of a model of the phenomena.

Response: We thank you for thinking our findings are novel and would be of interest to the community.

Is the work convincing, and if not, what further evidence would be required to strengthen the conclusions?

The work appears to be convincing, but there are some gaps in the presentation of the results and the subsequent discussion. Prior to publication, shortcomings mentioned in the major and minor issues sections should be addressed.

Response: We thank you very much for your valuable comments, which are great helpful to improve the quality of the manuscript. Detailed responses to the issues are given in the following. For the presentation of the results and discussions, considerable revisions have been made throughout the manuscript to clarify the working concept.

We would also be grateful if you could comment on the appropriateness and validity of any statistical analysis, as well as the ability of a researcher to reproduce the work, given the level of detail provided.

It is unclear how repeatable these results are between tests. Only one graph shows error bars in the whole paper, and the data for those error bars was taken from one test.

Response: First, we did more experiments (50+) and found the results are well repeated, based on which more experimental data are added (see revised Figs. 3b and 4). Second, we have designed two demos to show the efficiency of our adhesion modulation method (see **Videos 1 and 2** for details), which helps people to repeat our tests.

Figure 1 does a good job showing the experimental setup that is used to generate the results that are presented. However, more information on the brand or type of force sensor used, as well as the stiffness/material of the cantilever beam would be helpful to someone trying to recreate the study. The methods section should be updated to include parameters about the loudspeakers such as their size, impedance, resistance, and how the amplitudes of vibration were synced.

Response: We have revised the manuscript accordingly (refer to the revised **Methods**).

The description of the rubber band “..... with a rubber band buffer (with force constant of 121.4 N/m)” was updated as “..... with a rubber band buffer (cut from a yellow natural latex loop; section of $1.5 \times 1.5 \text{ mm}^2$, stiffness of 121.4 N/m)”.

The used commercial sensor is constructed by a cantilever beam with several strain gauges bonded in the surface, and no additional refitting design is introduced. Such sensor was used to measure the normal force near the free end (effect of the force location on the bend stress is removed automatically) when the other end is fixed. Thus, only the measurement range, resolution, and applicable frequency band are key parameters, and one can use any other sensor with similar parameters instead. In these respects, we updated the description of the force sensor “.....121.4 N/m) and a cantilever force sensor (range $\pm 2 \text{ N}$, resolution $30 \text{ }\mu\text{N}$)” as “.....121.4 N/m) and a force sensor. The force sensor (with measure range of $\pm 2 \text{ N}$, resolution of $30 \text{ }\mu\text{N}$, and working frequency $< 2 \text{ kHz}$) was constructed by a cantilever beam with several strain gauges adhered to the surface and which was used to measure the normal force near the free end”.

We have added the parameters about the loudspeakers in the revised manuscript: “Two electrodynamic coil cone loudspeakers (with diameter of 8 inch, rated power of 200 W, and impedance of $8 \text{ }\Omega$) were used to excite the vibration of the PDMS platform”.

For syncing the amplitudes of vibration, we added descriptions as “The loudspeakers were driven by a harmonic signal generator (with working frequency $< 20 \text{ kHz}$) equipped with a power amplifier (with working frequency $< 10 \text{ kHz}$ and power of 1 kW). The vibration amplitude was controlled by adjusting the amplification factor of the power amplifier”.

In addition, we added description that “Displacement data is recorded by a data acquisition card (NI USB-6212)”.

Major Issues

1. The study fails to address how the findings relate to previous research in this area. The authors should rewrite their introduction and discussion to reference the most closely related literature on adhesion for robotics, and clearly explain the relative

advantages and disadvantages of the proposed approach.

Response: The introduction and discussion have been rewritten in this revision. We believe that the revised version can address how the findings relate to previous research in this area.

2. The authors never directly show the forces they can achieve, but only show a relative increase or decrease in performance. However, it is important to present the base-line pull-off data, which is never explicitly presented or calculated. For a discussion of practicality, which is a main claim of this paper, it is important to discuss whether or not this system generates enough adhesion to be useful.

Response: Thank you for the great suggestion. In the revision, we have added a figure (see Figure below or revised Fig. 2a) to demonstrate the measured force data directly. In addition, to show the potential usefulness of our method, we have designed two demos to show its effectivity (see **Videos 1 and 2** in the supplementary materials for details).

3. Fig 2b: The wording used to describe this test is confusing, specifically pull-off. I assume this test was performed by using the quasi-static loading methodology discussed later in the paper to achieve a 70x improvement and then it was left to vibrate. If this is the case, full pull-off never occurs during this test. The only pull-off actually performed is for the non-vibrating case.

Response: Yes. We are sorry to misuse \bar{F} by F_{off} here. In fact, we meant that the vibration was kept on all the way, the contact load was increased slowly by moving the cantilever beam upwards by the vertical translation stage until to reach 70 times of $F_{\text{off},0}$, and then turn off (hold on) the translation stage till the end of the experiment (see the added “**Durability test**” in **Methods**). We replaced F_{off} in Fig. 2b (Fig. 2c

in the revision) and the description in the figure legend is revised as “c, A durability test by keeping the apparent (or average) contact load constant ($\bar{F} \approx 70F_{\text{off},0}$, point “o” in (a)), where the drastically enhanced adhesion strength is maintained for over 2×10^7 vibration cycles without any noticeable degradation in the adhesion performance”. The related contents in the main text have been rewritten (refer to the 4th paragraph of the main text).

4. Fig 3b: It is difficult to assess if the model is a good match as region one only possesses two data points. Please present some measure of the variation for each experimental trial and/or more data points.

Response: We have added a series of corresponding experiments (50+) and updated the data points in revised Fig. 3b (or Figure below), where the red circles are experimental data.

5. Fig 4: There are relatively few experimental samples shown for high frequency and high amplitude cases. Data should be presented for these cases or a rationale provided for why these tests were not performed. What was the sampling strategy for this set of experiments?

Response: Our original experiment scheme is let the data points uniformly cover the $A_b \sim \omega/(2\pi)$ map. However, the mass of the Support (under the PDMS platform in Fig. 1) was a little large before, and it consumed too much power. Thus, limited by the input power, the vibration amplitude was limited at higher frequencies. During the revision, we optimized the structure of the Support, and hence new experiments for high frequency and high amplitude cases covering the $A_b \sim \omega/(2\pi)$ map were performed (see revised Figs. 3b and 4 for details).

6. Fig S3: It is not always obvious when the test starts and when the test ends from inspecting the graphs. Please improve the clarity.

Response: We guess the question of the Referee is how to distinguish the switching time from the figure, i.e. by finding out the starting and ending time. Fig. S3 demonstrates a small number (specifically, in hundreds) of vibration cycles intercepted from the massive recorded cycles. The loops under one vibration state locate close to each other. Thus, two vibration states should have two clusters of loops. One can find two distinguishable clusters of loops in Fig. S3(g)-(n). Sometimes, the two clusters of loops may overlap with each other, e.g. in Fig. S3(a)-(f). But for all these cases, by careful analyzing the changing of curves with time, we can identify remarkable ungrouped loops during the adhesion switching process. Thus, number of the ungrouped loops (N) in Fig. S3 can be used to evaluate the switching time. We have added descriptions in **Section V** of the **Supplementary Information**.

7. In the supplemental materials, section I, the equation S1 is presented as being based on previous work, but the authors list 15 different citations in support of this claim. Is it true that all 15 papers directly address this claim? Perhaps the authors could be a bit more precise in their citations to guide the reader in their review of this previous work.

Response: Thank you for this valuable suggestion. We have written a more detailed and clear review about the previous work in **Section I** of the **Supplementary Information**.

8. Lines 7-8: The abstract claims two orders of magnitude improvement, but the reported data does not appear to support this conclusion.

Response: We are sorry for this misleading. In fact, what we were trying to say was that the regulation multiples ranges from 0.6x to 70x, which spans two orders of magnitudes. By carrying out new experiments, we find the enhanced adhesion strength can be 80 times while in the weakest case it is 0. We have revised the original description “The apparent adhesion strength can be varied by two orders of magnitude by controlling either the vibration frequency or amplitude” as “For typical PDMS-glass adhesion system, the apparent adhesion strength can be enhanced by 77 times or weakened to 0 by controlling either the vibration frequency or amplitude”.

9. Line 125: How was this verified? What observation did the authors make?

Response: In the experiments reflected by the revised Fig. 3b, we found that no further adhesion enhancement can be made when the vibration amplitude of the PDMS platform (A_b) is larger than $30\ \mu\text{m}$. Combining the recorded contact radius data (similar to Fig. 2b), we found that the maximum interfacial crack growth rate (identified by slope of the curve in Fig. 2b) is larger than v_c when $A_b > 30\ \mu\text{m}$.

10. Lines 150-151: The load is increased from 35x to 45x. What would it look like for a more dramatic change such as going from 7x to 70x?

Response: This is a very good suggestion. We have designed a new scheme to complete such an experiment and revised the original descriptions. The results are shown in updated Fig. S2(j)-(l). The description “Typical results are shown in Fig. S4 and Table S1, where the adhesion was tested under a sudden change in actuation amplitude (e.g. from $A_b = 30$ to $80\ \mu\text{m}$) or in actuation frequency (e.g. from $\omega/(2\pi) = 300$ to $500\ \text{Hz}$), and the adhesion switching was found to finish within $16\ \text{ms}$. Similarly, by an abruptly increasing of load impact from $F_{\text{off}}/F_{\text{off},0} = 37$ to 45 , the transformation time is also determined as small as $30\ \text{ms}$ ” is revised as “Typical results are shown in Fig. S4 and Table S1, where we measured the adhesion by a sudden change either in the actuation amplitude (e.g. from $A_b = 30$ to $80\ \mu\text{m}$) or in the apparent contact load (e.g. from $\bar{F}/F_{\text{off},0} = 6$ to 69) with a constant actuation frequency of $450\ \text{Hz}$, we observed that the system response very quick (within $16\ \text{ms}$, Fig. S4 and Table S1). Similarly, by an abruptly increasing of the actuation frequency from $\omega/(2\pi) = 300$ to $500\ \text{Hz}$ with a constant actuation amplitude of $30\ \mu\text{m}$, the transformation time is measured as small as $33\ \text{ms}$ (Table S1)”.

Minor Issues

1. The authors should generally revise the language of the manuscript to improve readability and clarity.

Response: Yes, careful revisions have been made throughout the manuscript to improve the quality of writing.

2. Fig 3a: Can the failure points indicated by the blue line be verified using

experiments?

Response: This is a good suggestion but it's not easy in practice. In recording the raw experimental data in Fig. 2b and Fig. 4, the storage space is taken up to 2 TB. For the records from the high-speed camera, we have to record for substantial amount of time until the two contact part pull off because it's impossible to predict exactly when it will pull off. Therefore, one need large storage space to identify one point on the blue line. As for identifying enough points on the blue line, based on our experience in recording the raw experimental data in Fig. 2a, the required storage space may be up to 100 TB. It seems to that the cost of required high performance RAID's can hardly be afforded. In addition, recording and processing these data will take more than four months. We believe that such an attempt is a good choice for follow-up research.

3. Line 64: Please cite or explain the source of this relation.

Response: Thank you for your reminding. The relation $K \approx 16E/9$ is derived from the relation $K = 4E^*/3$, where $E^* = E/(1 - \nu^2) \approx 4E/3$ considering the Poisson's ratio $\nu \rightarrow 0.5$. The relation $K = 4E^*/3$ can be identified from such as the JKR theory (refer to *Surface energy and the contact of elastic solids* by K. L. Johnson, K. Kendall and A. D. Roberts for example) since the Young's modulus of the glass is far larger than that of the PDMS. We have added a brief explanation and the citation in the revised manuscript.

4. Line 148: Reference to fig S4, but should be S3

Response: Many thanks to the Referee for carefully reading our manuscript. We have replaced the "Fig. S4" by "Fig. S3".

5. Table S2: Is the power presented in this table a theoretically or experimentally determined value? Please specify.

Response: The power in Table S2 is experimentally determined value. The original description "where $P/(\pi a^2)$ represents required input power per unit contact area" is updated as "where $P/(\pi r_{\text{eff}}^2)$ represents required input power per unit contact area and it is experimentally determined based on Eq. (S20) and corresponding recorded displacement data".

Responses to Reviewer #3

The authors introduce a method by which interfacial adhesion can be regulated by exciting mechanical “micro-vibration” perpendicular to the contact plane. They report that apparent adhesion strength can be varied by two orders of magnitude by control of frequency and amplitude of vibrations. Switching can be rapid (~ 15 ms) and repeated more than 107 times. The method requires no special microstructure. This could be a very important paper but several things about how the experiments were done are unclear to me and parts of the theoretical model need to be better explained. In addition, the outline of the theory needs some more physical explanation to explain the effect. So, I believe that the manuscript needs a significant revision before being reviewed again. To repeat, this is potentially an important and novel paper but, as written, I am unable to decide if that is the case. I would ask the authors to address the following points.

Response: Many thanks to the referee for considering our work is potentially important and novel. Your great comments have helps to significantly improve the manuscript.

1) Language needs some work.

Response: Careful revisions have been made throughout the manuscript to improve the quality of writing.

2) Some of the words in Figure 1 are very difficult to read.

Response: We have revised the captions and marks of Fig. 1 to make is easy to read.

3) I don't follow the caption of Figure 2b. It purports to show pull-off force for 2 x 107 cycles but at the same time it is stated that contact radius barely changed (Fig. 2a). I would like to see a full force-displacement curve to go with the Fig. 2a. Is it that the normal force went up transiently to 70 times the pull-off force without vibration? This is critical point to explain about how pull-off forces were measured. 4) Similarly in

the Methods section (near line 175), exactly how the pull-off force was measured is not at all clear. Again, it is important to show the measured force-displacement data so that we can witness the reported increase in pull-off force from this measurement.

Response: We sincerely thank you for your valuable comments. First, we added a new figure, i.e. Fig. 2a in the revised manuscript, to demonstrate how the pull-off force was measured. Second, we revised the misusing of \bar{F} by F_{off} in original Fig. 2b (More details can be found in Major Issue 3 from Review 2). Corresponding description can be found in the revised legend of Fig. 2 and **Methods**.

5) On line 54, the asymmetry of radius change could occur due to other reasons such as effect of rate on adhesion energy, bulk viscoelasticity, inertia, etc. How do the authors ascribe it to the difference between crack growth versus closure?

Response: We are sorry for this confusion because of our misleading description. We agree with the referee that the asymmetry of radius change could occur due to the mentioned reasons. But considering that experimentally the adhesion work is generally measured by the peeling test, where the external force and displacement are recorded (the input work can thus be calculated) as the crack grows. In principle, all of the mentioned factors could contribute to the adhesion work and thus should also be reflected in the crack growing or healing rate. This is why we ascribe the asymmetry of radius change to the difference between crack growth versus closure though essentially it results from either the rate effect of adhesion energy or bulk viscoelasticity, etc. To avoid the confusion, The sentence “which we attribute to the difference in the interfacial resistances between crack growing (with a decreasing) and healing (with a increasing), induced by the asymmetry of the effective adhesion work (r, v) (r is the Polar radius with the Pole at the contact center, $v \approx -\dot{a}$; see Fig. S1 and Eq. (S1) for details)” is updated as “which could originate from the effect of rate on the adhesion energy, bulk viscoelasticity, inertia, etc., reflected in the different crack growing (with a decreasing) and healing (with a increasing) rates.”

6) Line 64, to which elastic modulus are the authors referring (K)?

Response: As Reviewer 2 has similar query, we just repeat the response there. The relation $K \approx 16E/9$ is derived from the relation $K = 4E^*/3$, where $E^* = E/(1 - \nu^2) \approx 4E/3$ considering the Poisson's ratio $\nu \rightarrow 0.5$. The relation

$K = 4E^*/3$ can be identified from such as the JKR theory (refer to *Surface energy and the contact of elastic solids* by K. L. Johnson, K. Kendall and A. D. Roberts for example) considering that Young's modulus of the glass is far larger than that of the PDMS. We have added a brief explanation and the citation in the revised manuscript.

7) Equations (2) and (S1) are not the same. In SI, equation (S1) is introduced by referring to many papers. However, since this is a critical part of the paper, the physical basis for this interfacial constitutive law must be explained, at least in SI. For example, the model assumes that at very slow velocities the work of opening and closing the interface is the same, but this is usually not true for Sylgard 184 PDMS even at slow enough rates that the works of adhesion show no rate effect.

Response: Thank you for this valuable suggestion. We have added a more detailed and clear review about the previous work related to Eq. (S1), and descriptions about the physical basis of the interfacial constitutive law are also given. Details can be found in Section I of the revised **Supplementary Information**. In addition, we did not mean that “the model assumes that at very slow velocities the work of opening and closing the interface is the same”. In Fig. S1, the red and green curves seem overlap at very slow velocities, which is because the velocity is described by logarithmic coordinate. In fact, it is only assumed that the work of opening and closing the interface is the same only if $v = 0$, and Fig. S1 is in consistent with Eq. (S1) when $v < v_c$.

8) I) Please provide a reference for the Lagrange equations of motion for nonconservative systems. II) Please provide references to the form of elastic strain energy given on page 2/12 of SI. III) The specification in eq. (S1) that the first two equations apply only at $r=a$ is not clear to me and so it is also not clear because of this definition how the derivatives with respect to a in equation (S2) have been carried out. This needs to be clarified. IV) It is not clear how the red line in Figure S1 relates to the first part of eq. (S1), which does not tell us what happens for positive velocities greater than v_c .

Response: Thank you for your valuable comments.

I) For the Lagrange equations of motion for nonconservative systems, we referred a textbook *Analytical Mechanics* (In Chinese, Tsinghua University Press) by Yonggang

Wang. We have added the citation in this revision.

II) For the form of elastic strain energy given on page 2/12 of Eq. (S1), we have listed three references as book *Contact Mechanics and Friction: Physical Principles and Applications* by V.L. Popov, book *Contact Mechanics* by K. L. Johnson, and the famous article *Surface energy and the contact of elastic solids* by K. L. Johnson, K. Kendall and A. D. Roberts.

III) We revised the derivation of Eq. (S2) as: “Without losing of generality, we considered a fixed base and a vibrating contactor (Fig. 1b), the kinetic energy of the contactor is $E_k \approx M\dot{\delta}^2/2$, the elastic potential energy of the system is $E_p \approx 3/4 \cdot K(\delta^2 a - 2/3 \cdot \delta a^3/R + a^5/R^2/5)$ ³³⁻³⁵, the bulk dissipation function is $D = c\dot{\delta}^2/2$ (c is the bulk damping coefficient), the generalized force with respect to the generalized coordinate δ is $Q_\delta = F = \bar{F} + f$, and the generalized force with respect to the generalized coordinate a is $Q_a = 2\pi a w(a, v)$, deduced from the elementary work of adhesion $dW_A = 2\pi a w(a, v) da$. According to the Lagrange equations for non-conservative systems³⁶, i.e.

$$\begin{cases} \frac{d}{dt} \frac{\partial E_k}{\partial \dot{\delta}} - \frac{\partial E_k}{\partial \delta} + \frac{\partial E_p}{\partial \delta} + \frac{\partial D}{\partial \dot{\delta}} = Q_\delta, \\ \frac{d}{dt} \frac{\partial E_k}{\partial \dot{a}} - \frac{\partial E_k}{\partial a} + \frac{\partial E_p}{\partial a} + \frac{\partial D}{\partial \dot{a}} = Q_a, \end{cases} \quad (\text{S2})$$

one can obtain the governing equation,

$$\begin{cases} \frac{Ka}{2} \left(3\delta - \frac{a^2}{R} \right) + M\ddot{\delta} + c\dot{\delta} = \bar{F} + f, \\ \frac{3K}{8\pi a} \left(\delta - \frac{a^2}{R} \right)^2 = w(a, v) = \begin{cases} w_0 \left(1 + \left| \frac{v}{v_0} \right|^\alpha \right), & 0 \leq v \leq v_c, \\ w_0 \left(1 + \left| \frac{v}{v_0} \right|^\alpha \right)^{-1}, & -\infty < v \leq 0, \end{cases} \end{cases} \quad (\text{S3})$$

”.

IV) The peeling behavior when $v > v_c$ is a very interesting issue. As far as we know, it is still an open question. We believe that more in-depth discussion of such issue

goes far beyond the scope of this study. We have planned to study the issue in future research.

9) It seems that a and δ being treated as independent kinematic quantities? In contact mechanics they are usually related.

Response: Yes, if considering the static equilibrium state, a and δ are related as (refer to the JKR theory) $(3\delta - a^2/R)Ka/2 = F$ and $3K/8\pi a(\delta - a^2/R)^2 = w_0$. However, at the dynamic state, such relations become dynamically determined, i.e. there is no uniquely determined relations between a and δ if the dynamic effect is unknown. This is the typical characteristics of a dynamic system with two free degrees, and one should treat them as two generalized coordinates to solve the dynamic problem when the method of analytical mechanics is used.

10) The claim in line 160 that the authors have demonstrated a method for modulating adhesion that works for “any adhesive materials (sic)” is too broad. Their demonstration depends on a particular rate dependence of adhesion and the effect has been shown only for one materials system.

Response: We have updated the description “... of any adhesive materials” as “... of any **materials with rate-dependent adhesion**”. On the other hand, we added a demo in this revision (refer to **Section VII** of the **Supplementary Information** and **Videos 1** and **2**), which used another materials system “PV-glass” instead of the original “PDMS-glass” to demonstrate generality of the method. In fact, rate dependence of adhesion is very common in adhesive materials.

11) I do not follow how the authors get S4 from S3. There is some approximation involved that has not been stated, it seems. Please clarify.

Response: The difficulty in understanding may due to some typos in the original description “ ε can be assumed far less than a_0 ” in the first paragraph of Section II of the Supporting materials. Some symbols were used in earlier versions and forgotten to be modified during modifications. In fact, “ ε ” should be “ $|\varepsilon|$ ”, and “ a_0 ” should be “ $|\bar{a}|$ ”. The original description “ ε can be assumed far less than a_0 ” has been revised as “ $|\varepsilon|$ is far less than \bar{a} ”. In this case, Eq. (S4a) and the left hand of Eq. (S4b) can be directly obtained just by $a \approx \bar{a}$ because $|\varepsilon|$ is far less than $|\bar{a}|$. It is important to

note that $\dot{a} = \dot{\varepsilon}$ (the upper dot means derivation with respect to t) cannot be ignored because “ ε is a periodic function with respect to ωt ”. Considering $v = -\dot{a} = -\dot{\varepsilon}$, one can obtain the right hand of Eq. (S4b).

12) The authors quote but don't discuss the paper by Wahl et al. (ref. 23) which examined a closely related experiment in which I don't believe any significant enhancement of pull-off force was reported. The authors need to compare their work with that of ref 23.

Response: There are two references about the work:

[A] Greenwood, J.A. & Johnson, K.L. Oscillatory loading of a viscoelastic adhesive contact. *J Colloid Interf Sci* **296**, 284-291 (2006).

[B] Wahl, K.J., Asif, S.A.S., Greenwood, J.A. & Johnson, K.L. Oscillating adhesive contacts between micron-scale tips and compliant polymers. *J Colloid Interf Sci* **296**, 178-188 (2006).

There is no significant enhancement of pull-off force in the experiment by Wahl et al. indeed. We have estimated the pull-off force in their experiments based on our theory (see below for details), our conclusion is that their actuation frequency/amplitude is not enough to regulate the adhesion.

Note that $a_r \bar{a}$ and $a_r^2 \bar{\delta}/R$ in ref. A are equal to a and δ in our work, respectively, where $a_r = \sqrt[3]{9\pi R^2 w_0 / (4E^*)}$ (ref. A), $R \sim 10 \mu\text{m}$ (ref. B), $w_0 \sim 40 \text{ mJ/m}^2$ (this work), and $E^* = E / (1 - \nu^2) \approx 4E/3 \sim 5 \text{ MPa}$ (refer to ref. 62 of the main text), and \bar{a} and $\bar{\delta}$ are average values of a and δ in our work, respectively. Based on the Fig. 6 in ref. A (or Figure below), one can distinguish non-dimensional contact radius and penetration depth as ~ 0.998 and ~ 0.05 on average, respectively. By using Eq. (S5) in our **Supplementary Information** $\bar{F} = (3\bar{\delta} - \bar{a}^2/R) K \bar{a} / 2$, we have $\bar{a} = 0.998 a_r = 1.778 \mu\text{m}$, $\bar{\delta} = 0.05 a_r^2 / R = 15.87 \text{ nm}$, $K = 16E/9 = 6.667 \text{ MPa}$, and finally $\bar{F} = -1.591 \mu\text{N}$. On the other hand, the JKR pull-off force is $F_{\text{off},0} = -3\pi R w_0 / 2 = -1.583 \mu\text{N}$. It is obvious that $\bar{F} \sim F_{\text{off},0}$, i.e. no significant adhesion enhancement can be achieved in their experiments.

Fig. 6. (a) Variation of radius $\bar{a}(W)$ during load modulation. The loci are closely elliptical. (b) Variation of displacement $\delta(W)$ during load modulation.

13) For these reasons, I have been unable to check all the equations in SI.

Response: We have tried our best to clarify the equations deduction.

14) Some more detailed and better physical explanation for this effect is needed.

Response: Yes, thank you for your valuable suggestion. We have made a series of revisions throughout the manuscript to clarify the working concept and demonstrate better physical explanation (see also our Response to Referees 1 and 2 above).

Reviewers' comments:

Reviewer #1 (Remarks to the Author):

This paper describes a method for modulating the adhesion between a suspended glass indenter and an adhesive substrate via vibration of the substrate. Depending on the frequency and amplitude of vibration, adhesion can be increased substantially above the static case ($\sim 70\times$) and slightly below ($0.6\times$). A model describing this behavior is presented and data is shown that supports the model. The time for switching between high and low adhesion is tested and shown to be around 15-30ms. The authors claim potential applications in gecko-inspired climbing robots.

While this work presents a very intriguing concept (vibration-modulated adhesion), this reviewer has major concerns that would need to be addressed before publication.

Response: We thank you very much for thinking our work very intriguing.

First, it is unclear whether the applicability of the concept is as broad as proposed. The system that is tested and modeled has a spherical indenter suspended by a rubber band, through which load is applied, while the adhesive substrate is vibrated. Translating this to a practical system might not be trivial. For instance, for a climbing robot, would the vibration element be contained in the foot of the robot? Is that practical? Will the flat-flat contact that would be expected in a climbing robot behave the same? Would the high shear loads (the primary load for a climbing robot) affect the results? To this reviewer, either these questions need to be addressed or the emphasis on this application should be removed. Ideally, the authors would show a simple demonstration of an adhesive pad suspending different masses depending on vibration characteristics. Further, only a small change in adhesion is seen between the default adhesion and a reduced adhesion state, meaning to make a "switchable" adhesive, the vibration would have to be continuously on for the "strong" state. This could be a limitation for robots that are designed to remain passively on the wall for long periods of time.

Response: We thank you very much for these constructive suggestions. They are great helpful to improve the value and quality of the work. To address your questions, we have added a simple demo (refer to Video 1 in the supplementary materials) as suggested. In the demo: (a) The vibration elements (here are small converted electromagnetic loudspeakers) are contained in the contactors. In such scheme, the vibration element is contained in the "foot" of the "robot". (b) It is almost a flat-flat contact (the contactor has in-plane diameter of 15 mm and out-of-plane curvature radius of 80 mm, i.e. height difference between edge and center of the contactor is about 0.35 mm). Specifically, three contactors covered by commercial thin PVC anti-slip mats are fixed to a triangular plastic frame. When turn on the vibration elements, the plastic frame with the contactors adheres to a flat glass wall well, while once turn off the vibration elements, the plastic frame falls by gravity (see Video 1 for details). To study the effects of shear loads on the adhesion, we simply rotate the adhesion system in Video 1 by 90 degrees, which mimics the climbing robots, we found that the micro-vibration can still modulate the interface adhesion well though shear loads (due to the gravity of the "robot" in Video 2) exerted on the interface.

=> This reviewer thanks the authors for a significant effort to address this concern. It is very helpful to see these demonstrations. They go a long way to inform the reader of the practicality of the concept.

As for the comments of "only a small change in adhesion is seen between the default and a reduced adhesion state", we have added new experiments to show significant changes in adhesion (see revised Figs. 3b and 4), the adhesion strength can be reduced to 0% of the default/quasi-static/intrinsic one.

=> These new data are helpful. Reducing to 0% is a good result.

As the limitation of “the vibration would have to be continuously on for the ‘strong’ state” commented by the referee, we admit that, but fortunately, we found that the system has very low power consumption (~100 mW/mm², refer to the 2nd to last paragraph of the main text), and thus it is expected to be competent for most of applications.

=>While this is helpful, please replace the text that says “this is very low,” and instead provide a comparison to other energy requirements of a climbing robot. The statement “this is very low” is not concrete. For instance, you could estimate the area needed for a 1kg robot and predict the power consumption, then note the power required for the robot to climb at a given speed (or the power consumed by electronics in an idle state, or possibly the power required to transmit information wirelessly).

Second, the purported impact on climbing robots and gecko-inspired adhesives is likely overstated, given the lack of appropriate references. It is claimed “Adhesion switching has been proposed by either altering the load path and failure mode of the interface to form a “smart/phase change” interface or controlling the debonding speed, yet only two adhesion states, i.e. “strong” or “weak”, can be identified and the switching speed is intrinsically limited.” This misses a whole class of adhesives, known as “controllable,” inspired by the gecko [for instance: R1, R2, R3, R4]. The normal adhesive force can be continuously tuned by the direction of the applied load.

Further, these adhesives have been shown to switch on the order 20ms, roughly the same rate as reported in the presented work [R5].

Response: Thank you for suggesting these important references. We have added them to our References. We completely agree with the actual characteristics of the class of adhesives that “the normal adhesive force can be continuously tuned by the direction of the applied load”. The suggested references have been added to the manuscript as ref. 5, 13-16. In addition, the original description about “Adhesion switching” has been revised as (refer to 2nd paragraph of the main text): “Considering that the apparent adhesion strength can be affected by the real contact area, the size of contact region 17, and the adhesion work 20, adhesion switching has been proposed by either controlling the load path and failure mode of the interface or constructing a “smart” interface such as phase transition interface 6, 20, based on which diversity of adhesion switching triggers e.g. mechanical 12, 15, 21-30, electro/magnetic 14, 16, 31, lighten 32-34, fluidic 35-38, thermal 39-44, have been reported. Note that the gecko-inspired directional adhesion 12, 13, 15, 16, 45-50 and the debonding/peeling speed-regulated adhesion 24-30 provide the promising methods to tune the adhesion strength continuously and rapidly. However, at present, there is still lacking of simple and long-term effective adhesion modulation method. In fact, in most cases, only two adhesion states, i.e. “strong” or “weak”, can be identified, or easy to de-adhere but hard to re-adhere rapidly, or the switching speed is intrinsically limited 6, 20.”

=>Thank you for adding these citations. However, the current text is still misleading, in this reviewer’s opinion, and should be corrected. It is stated: “However, at present, there is still lacking of simple and long-term effective adhesion modulation method. In fact, in most cases, only two adhesion states, i.e. “strong” or “weak”, can be identified, or easy to de-adhere but hard to re-adhere rapidly, or the switching speed is intrinsically limited 6, 20.” If one reads the new citations, this is simply not true. These references describe “controllable” adhesives that behave like a gecko’s adhesive. They show a simple and long-term effective adhesion modulation method. They are not either “strong” or “weak,” but can be continuously varied from zero to maximum adhesion depending on the applied shear load. And their switching speed is not intrinsically limited down to 20ms.

Third, the work is not presented clearly. The most important part of the work is a simple and clear explanation of why this effect occurs. However, besides presentation of data showing the asymmetry of growth and retraction of the contact patch, a clear explanation is not provided. In a single sentence, why does this work? More generally, the organization and presentation are not clear. This is in

addition to consistent English errors throughout.

Response: We have revised our manuscript to make the presentation clear and tried our best to correct the English errors. In a single sentence, the proposed method works because the interfacial adhesion work significantly and asymmetrically depends on the speed of the moving crack tip (the edge of the contact interface), based on which we introduce a micro-vibration to the adhesion system to effectively modulate the moving speed of crack tip.

=>This is close. However, this reviewer thinks it could still be stated more clearly. Can you explicitly state how the asymmetry in interfacial adhesion work depends on speed and how this allows you to enhance adhesion in some cases and decrease it in others? Critically, this needs to be introduced earlier. Please put it in the abstract. Right now, the reader has no idea about your main contribution until somewhere deep in the fourth paragraph.

Overall, while this work is intriguing, it is not ready yet for publication in a journal of the caliber of Nature Communications. A much tighter, more clear, well crafted manuscript is needed that carefully considers the proposed applications and explains simply the working concept.

Response: We have rewritten the introduction and the discussion parts. Considerable revisions have been made throughout the manuscript to clarify the working concept. We believe the revised version has been significantly improved based on the referee's comments/suggestions.

=>The writing is still not at the level suitable for Nature Communications. There are typos throughout, especially in the newly added material.

=>Finally, there are two new concerns that needs serious consideration.

1) There are a series of wall climbing adhesive robots that use vibration to enhance the adhesion of their rubber feet equipped with suction cups. While the presented work clearly makes a contribution with respect to this previous work, it needs to be cited and mentioned, since the basic concept is shared by both works: enhancing adhesion through vibration by utilizing adhesion hysteresis, where jump-in to adhesion is less strong than pull-off.

-Zhu, T., Liu, R., Wang, X. D., & Wang, K. (2006, December). Principle and application of vibrating suction method. In 2006 IEEE International Conference on Robotics and Biomimetics (pp. 491-495). IEEE.

-He, Na Shun Bu, Rong Liu, Wei Wang, Hao Yang, and Xu Dong Wang. "A mini multi-joint wall climbing robot based on the vibrating suction method." In 2007 IEEE International Conference on Robotics and Biomimetics (ROBIO), pp. 1861-1865. IEEE, 2007.

-Liu, Rong, Qingfeng Hong, and Na Shun Bu He. "A miniature multi-joint wall-climbing robot based on new vibration suction robotic foot." In 2008 IEEE International Conference on Automation and Logistics, pp. 1160-1165. IEEE, 2008.

2) The advantages and disadvantages of the current technology compared to that already used for climbing robots is needed. This does not mean you simply say what others are doing then later say what you did. You need to directly compare the pros and cons. For instance, your concluding sentence should state what your solution adds compared to the state of the art, but currently it could be describing already-published controllable adhesion work: "Moreover, we find that the adhesion switching by our method is very quick, on the order of 10 ms, comparable to that of geckos (15 ms). Considering the simple and reliable adhesion modulation together with the intrinsically free of preload procedure, our method could be promising for the application in intelligent adhesion contact system." Previously published work on controllable adhesion shows fast switching, is a simple and reliable modulation method, and is intrinsically free of preload procedures. This reviewer believes your work does add to the state-of-the-art, but it is not these items, since they have already been done.

Reviewer #2 (Remarks to the Author):

Overall I feel that there is substantial amount of work done with impressive results. The authors have addressed most of the reviewer comments and concerns. The questions that remains unaddressed are as follows:

1. Although the results are very impressive and the concept is novel, the underlying physics that causes this phenomenon is still unclear to me. The authors have tried to explain it in text. However, having schematics to explain the concept of micro-vibration based adhesion will make the underlying concept easy to understand.

2. It is unclear what kind of statistical approach was used for post processing the results for the experiments. Were the results obtained from the same single sample with multiple trials or multiple identical samples with sets of trials? What is the confidence interval for the error bars? What was the standard deviation for the experiments? More details about the statistical approach used for post processing would be needed to evaluate the quality of results.

Also, there are a few grammatical mistakes that need to be corrected before the final submission.

Reviewer #3 (Remarks to the Author):

I am mostly satisfied with the authors' response to the reviewers' comments. I think this is a novel and interesting discovery and deserves to be published in Nature Communications. However, two important items remain to be addressed before publication. I don't think the manuscript needs to be reviewed again except by the editors.

1) Overall, the language needs considerable work for readability, clarity, and to remove typographical errors.

2) While the dynamical model is now better described, and the authors have made some attempt in their response to reviewers to give a physical explanation of their effect, this has still not made it into the main manuscript. It will take only a few lines but I think it is imperative for the authors to describe clearly the physical effect that makes vibration of chosen frequency and amplitude modulate the friction dramatically.

Responses to Reviewer #1

This paper describes a method for modulating the adhesion between a suspended glass indenter an adhesive substrate via vibration of the substrate. Depending on the frequency and amplitude of vibration, adhesion can be increased substantially above the static case (~70x) and slightly below (0.6x). A model describing this behavior is presented and data is shown that supports the model. The time for switching between high and low adhesion is tested and shown to be around 15-30ms. The authors claim potential applications in gecko-inspired climbing robots.

While this work presents a very intriguing concept (vibration-modulated adhesion), this reviewer has major concerns that would need to be addressed before publication.

Response: We thank you very much for thinking our work very intriguing.

First, it is unclear whether the applicability of the concept is as broad as proposed. The system that is tested and modeled has a spherical indenter suspended by a rubber band, through which load is applied, while the adhesive substrate is vibrated. Translating this to a practical system might not be trivial. For instance, for a climbing robot, would the vibration element be contained in the foot of the robot? Is that practical? Will the flat-flat contact that would be expected in a climbing robot behave the same? Would the high shear loads (the primary load for a climbing robot) affect the results? To this reviewer, either these questions need to be addressed or the emphasis on this application should be removed. Ideally, the authors would show a simple demonstration of an adhesive pad suspending different masses depending on vibration characteristics. Further, only a small change in adhesion is seen between the default adhesion and a reduced adhesion state, meaning to make a “switchable” adhesive, the vibration would have to be continuously on for the “strong” state. This could be a limitation for robots that are designed to remain passively on the wall for long periods of time.

Response: We thank you very much for these constructive suggestions. They are great helpful to improve the value and quality of the work. To address your questions, we have added a simple demo (refer to Video 1 in the supplementary materials) as suggested. In the demo: (a) The vibration elements (here are small converted electromagnetic loudspeakers) are contained in the contactors. In such scheme, the vibration element is contained in the “foot” of the “robot”. (b) It is almost a flat-flat contact (the contactor has in-plane diameter of 15 mm and out-of-plane curvature radius of 80 mm, i.e. height difference between edge and center of the contactor is about 0.35 mm). Specifically, three contactors covered by commercial thin PVC anti-slip mats are fixed to a triangular plastic frame. When turn on the vibration elements, the plastic frame with the contactors adheres to a flat glass wall well, while once turn off the vibration elements, the plastic frame falls by gravity (see Video 1 for details). To study the effects of shear loads on the adhesion, we simply rotate the adhesion system in Video 1 by 90 degrees, which mimics the climbing robots, we found that the micro-vibration can still modulate the interface adhesion well though shear loads (due to the gravity of the “robot” in Video 2) exerted on the interface.

Remark: =>This reviewer thanks the authors for a significant effort to address this concern. It is very helpful to see these demonstrations. They go a long way to inform the reader of the practicality of the concept.

Response: We thank you very much for your approval.

As for the comments of “only a small change in adhesion is seen between the default and a reduced adhesion state”, we have added new experiments to show significant changes in adhesion (see revised Figs. 3b and 4), the adhesion strength can be reduced to 0% of the default/quasi-static/intrinsic one.

Remark: =>These new data are helpful. Reducing to 0% is a good result.

Response: We thank you very much for your evaluation. We have highlighted this results in the revised manuscript.

As the limitation of “the vibration would have to be continuously on for the ‘strong’ state” commented by the referee, we admit that, but fortunately, we found that the system has very low power consumption (~100 mW/mm², refer to the 2nd to last paragraph of the main text), and thus it is expected to be competent for most of applications.

Remark: =>While this is helpful, please replace the text that says “this is very low,” and instead provide a comparison to other energy requirements of a climbing robot. The statement “this is very low” is not concrete. For instance, you could estimate the area needed for a 1kg robot and predict the power consumption, then note the power

required for the robot to climb at a given speed (or the power consumed by electronics in an idle state, or possibly the power required to transmit information wirelessly).

Response: We have deleted “this is very low” in the revised manuscript since the comparison with other types of adhesion system is not straightforward.

Second, the purported impact on climbing robots and gecko-inspired adhesives is likely overstated, given the lack of appropriate references. It is claimed “Adhesion switching has been proposed by either altering the load path and failure mode of the interface to form a “smart/phase change” interface or controlling the debonding speed, yet only two adhesion states, i.e. “strong” or “weak”, can be identified and the switching speed is intrinsically limited.” This misses a whole class of adhesives, known as “controllable,” inspired by the gecko [for instance: R1, R2, R3, R4]. The normal adhesive force can be continuously tuned by the direction of the applied load.

Further, these adhesives have been shown to switch on the order 20ms, roughly the same rate as reported in the presented work [R5].

Response: Thank you for suggesting these important references. We have added them to our References. We completely agree with the actual characteristics of the class of adhesives that “the normal adhesive force can be continuously tuned by the direction of the applied load”. The suggested references have been added to the manuscript as ref. 5, 13-16. In addition, the original description about “Adhesion switching” has been revised as (refer to 2nd paragraph of the main text): “Considering that the apparent adhesion strength can be affected by the real contact area, the size of contact region 17, and the adhesion work 20, adhesion switching has been proposed by either controlling the load path and failure mode of the interface or constructing a “smart” interface such as phase transition interface 6, 20, based on which diversity of adhesion switching triggers e.g. mechanical 12, 15, 21-30, electro/magnetic 14, 16, 31, lighten 32-34, fluidic 35-38, thermal 39-44, have been reported. Note that the gecko-inspired directional adhesion 12, 13, 15, 16, 45-50 and the debonding/peeling speed-regulated adhesion 24-30 provide the promising methods to tune the adhesion strength continuously and rapidly. However, at present, there is still lacking of simple and long-term effective adhesion modulation method. In fact, in most cases, only two adhesion states, i.e. “strong” or “weak”, can be identified, or easy to de-adhere but hard to re-adhere rapidly, or the switching speed is intrinsically limited 6, 20.”

Remark: =>Thank you for adding these citations. However, the current text is still misleading, in this reviewer’s opinion, and should be corrected. It is stated: “However, at present, there is still lacking of simple and long-term effective adhesion modulation method. In fact, in most cases, only two adhesion states, i.e. “strong” or “weak”, can be identified, or easy to de-adhere but hard to re-adhere rapidly, or the switching speed is intrinsically limited 6, 20.” If one reads the new citations, this is simply not true. These references describe “controllable” adhesives that behave like a gecko’s adhesive. They show a simple and long-term effective adhesion modulation method. They are not either “strong” or “weak,” but can be continuously varied from zero to maximum adhesion depending on the applied shear load. And their switching speed is not intrinsically limited down to 20ms.

Response: Thank you very much for your kind reminding. we have revised in the manuscript as “However, at present, there have been only a few attempts to provide promising methods to tune the adhesion strength continuously and rapidly, e.g., gecko-inspired directional adhesion^{12, 13, 15, 16, 45-50} and debonding/peeling speed-regulated adhesion²⁴⁻³⁰”.

Third, the work is not presented clearly. The most important part of the work is a simple and clear explanation of why this effect occurs. However, besides presentation of data showing the asymmetry of growth and retraction of the contact patch, a clear explanation is not provided. In a single sentence, why does this work? More generally, the organization and presentation are not clear. This is in addition to consistent English errors throughout.

Response: We have revised our manuscript to make the presentation clear and tried our best to correct the English errors. In a single sentence, the proposed method works because the interfacial adhesion work significantly and asymmetrically depends on the speed of the moving crack tip (the edge of the contact interface), based on which we introduce a micro-vibration to the adhesion system to effectively modulate the moving speed of crack tip.

Remark: =>This is close. However, this reviewer thinks it could still be stated more clearly. Can you explicitly state how the asymmetry in interfacial adhesion work

depends on speed and how this allows you to enhance adhesion in some cases and decrease it in others? Critically, this needs to be introduced earlier. Please put it in the abstract. Right now, the reader has no idea about your main contribution until somewhere deep in the fourth paragraph.

Response: We have revised the **Abstract** and 5th paragraph to state the mechanism. Specifically, the revised **Abstract** are: "...Here we show a robust and predictable method ... perpendicular to the contact plane. We observed that the introduced micro-vibration leads to the contact interface experiencing rapid cracking and healing within one cycle. Based on the fact that the cracking and healing of an interface generally possess different rate dependences, we developed an analytic model to reveal the underlying mechanism, which intrinsically originates from adhesion hysteresis and dynamic instability. For typical PDMS-glass adhesion...".

In the 5th paragraph, the sentence (before Eq. (2)) " $w(a, v)$ is the effective adhesive work at the contact line and refers to Fig. S1 and Eq. (S1) for its dependence on v ($v \approx -\dot{a}$)" has been revised as " $w(a, v)$ is the effective adhesive work at the contact line and significantly and asymmetrically depends on the speed of the moving crack tip v ($v \approx -\dot{a}$). Specifically, as the crack propagation speed (v) increases, the adhesion work first increases exponentially, then falls sharply and finally rises rapidly again. As the healing speed ($|v|$) increases, the adhesion work rapidly approaches zero (refer to Fig. S1)".

Overall, while this work is intriguing, it is not ready yet for publication in a journal of the caliber of Nature Communications. A much tighter, more clear, well crafted manuscript is needed that carefully considers the proposed applications and explains simply the working concept.

Response: We have rewritten the introduction and the discussion parts. Considerable revisions have been made throughout the manuscript to clarify the working concept. We believe the revised version has been significantly improved based on the referee's comments/suggestions.

Remark: =>The writing is still not at the level suitable for Nature Communications. There are typos throughout, especially in the newly added material.

Response: We have used Springer Nature Language Editing service to improve the clarity and readability.

Other two remarks: =>Finally, there are two new concerns that needs serious consideration.

1) There are a series of wall climbing adhesive robots that use vibration to enhance the adhesion of their rubber feet equipped with suction cups. While the presented work clearly makes a contribution with respect to this previous work, it needs to be cited and mentioned, since the basic concept is shared by both works: enhancing adhesion through vibration by utilizing adhesion hysteresis, where jump-into adhesion is less strong than pull-off.

-Zhu, T., Liu, R., Wang, X. D., & Wang, K. (2006, December). Principle and application of vibrating suction method. In 2006 IEEE International Conference on Robotics and Biomimetics (pp. 491-495). IEEE.

-He, Na Shun Bu, Rong Liu, Wei Wang, Hao Yang, and Xu Dong Wang. "A mini multi-joint wall climbing robot based on the vibrating suction method." In 2007 IEEE International Conference on Robotics and Biomimetics (ROBIO), pp. 1861-1865. IEEE, 2007.

-Liu, Rong, Qingfeng Hong, and Na Shun Bu He. "A miniature multi-joint

wall-climbing robot based on new vibration suction robotic foot." In 2008 IEEE International Conference on Automation and Logistics, pp. 1160-1165. IEEE, 2008.

Response: Thank you for suggesting these important references. They presented a suction-to-wall method, with which the negative pressure in the suction cup was established by high frequency vibration of the cup with respect to the wall surface. We have added a discussion after Eq. (3) as: "Eq. (3) indicates that F_{off} is higher than $F_{\text{off},0} = -3/2 \cdot \pi R w_0$ ($F_{\text{off}} = F_{\text{off},0}$ when $A_f = 0$) under vibration, i.e., $A_f > 0$, which can be attributed to the adhesion hysteresis induced by adhesion asymmetry as stated above. In fact, a similar "vibration suction method"⁶³⁻⁶⁵ has been reported to enhance the negative pressure through vibration."

2) The advantages and disadvantages of the current technology compared to that already used for climbing robots is needed. This does not mean you simply say what others are doing then later say what you did. You need to directly compare the pros and cons. For instance, your concluding sentence should state what your solution adds compared to the state of the art, but currently it could be describing already-published controllable adhesion work: "Moreover, we find that the adhesion switching by our method is very quick, on the order of 10 ms, comparable to that of geckos (15 ms). Considering the simple and reliable adhesion modulation together with the intrinsically free of preload procedure, our method could be promising for the application in intelligent adhesion contact system." Previously published work on controllable adhesion shows fast switching, is a simple and reliable modulation method, and is intrinsically free of preload procedures. This reviewer believes your work does add to the state-of-the-art, but it is not these items, since they have already been done.

Response: Thank you very much for thinking our work does add to the state-of-the-art. The sentence "in contrast to the prevalent adhesion controlling strategies" in the Abstract is deleted. In addition, we have revised the concluding sentences as: "Comparing the experimental results and theoretical model, we found that the adhesion can be either strengthened or weakened by controlling the micro-vibration, and the adhesion strength can be maintained at any desired value with good durability and reversibility within the theoretically permissible regulation range. Moreover, the adhesion switching occurs very quickly with our method, on the order of 10^1 ms, comparable to that of geckos (15 ms^4). We anticipate that our findings will pave the way for understanding and applying micro-vibration based interface adhesion regulation in intelligent adhesion technology"

Responses to Reviewer #2

Overall I feel that there is substantial amount of work done with impressive results. The authors have addressed most of the reviewer comments and concerns. The questions that remains unaddressed are as follows:

1. Although the results are very impressive and the concept is novel, the underlying physics that causes this phenomenon is still unclear to me. The authors have tried to explain it in text. However, having schematics to explain the concept of micro-vibration based adhesion will make the underlying concept easy to understand.

Response: Thank you for this valuable comment. We have tried to explain the concept by schematics. But it seems not easy to reach an ideal level that thoroughly improve the description. Compromisly, we revised the **Abstract** and 5th paragraph to state the mechanism. Specifically, the added sentences in the Abstract are: “Here we show a robust and predictable method ... perpendicular to the contact plane. We observed that the introduced micro-vibration leads to the contact interface experiencing rapid cracking and healing within one cycle. Based on the fact that the cracking and healing of an interface generally possess different rate dependences, we developed an analytic model to reveal the underlying mechanism, which intrinsically originates from adhesion hysteresis and dynamic instability. For typical PDMS-glass adhesion...”. In the 5th paragraph, the sentence (before Eq. (2)) “ $w(a, v)$ is the effective adhesive work at the contact line and refers to Fig. S1 and Eq. (S1) for its dependence on v ($v \approx -\dot{a}$)” is revised as “ $w(a, v)$ is the effective adhesive work at the contact line and significantly and asymmetrically depends on the speed of the moving crack tip v ($v \approx -\dot{a}$). Specifically, as the crack propagation speed (v) increases, the adhesion work first increases exponentially, then falls sharply and finally rises rapidly again. As the healing speed ($|v|$) increases, the adhesion work rapidly approaches zero (refer to Fig. S1)”

2. It is unclear what kind of statistical approach was used for post processing the results for the experiments. Were the results obtained from the same single sample with multiple trials or multiple identical samples with sets of trials? What is the confidence interval for the error bars? What was the standard deviation for the experiments? More details about the statistical approach used for post processing would be needed to evaluate the quality of results.

Response: Thank you very much for this constructive comment. The details are as following.

First, a rough scan on one sample is conducted to find out a group of typical frequency and amplitude exhibiting relatively good performance. The frequency and amplitude are identified as ~450 Hz and ~100 μm , respectively.

Second, we set 5 group of similar samples and consider 3 different contact points in each sample (i.e. totally there are 15 parallel trials) to test the performance under the typical frequency 450 Hz and amplitude 100 μm . We compared the pull-off forces of all trials ($F_{\text{off}}/F_{\text{off},0}$: 78.02, 78.31, 77.89, 75.37, 75.00, 75.48, 77.26, 77.21, 76.96, 66.14, 72.37, 69.60, 80.11, 79.82, 79.44). The mean, standard deviation and standard errors of the pull-off forces are 75.93, 3.90 and 1.01, respectively.

Third, the same single sample (we changed the contact point for several times because

of interfacial failure) is used to conduct all following formal tests, and the data embodied in the manuscript are all from these tests.

There are notes for the error bars in Fig. 2b. The data are collected from three consecutive vibration cycles in one trial, and each cycle contains 7 data points. The bars denote the standard errors (S.E., stated in Fig. 2b) which is calculated by

$S.E. = \sqrt{\frac{\sum_{i=1}^n (x_i - \bar{x})^2}{n(n-1)}}$, where $n = 3$. The bar means the interval $[\bar{x} - S.E., \bar{x} + S.E.]$.

More detailed data supporting the findings of this manuscript are available from the corresponding authors upon reasonable request.

Also, there are a few grammatical mistakes that need to be corrected before the final submission.

Response: We have used Springer Nature Language Editing service to improve the clarity and readability.

Responses to Reviewer #3

I am mostly satisfied with the authors' response to the reviewers' comments. I think this is a novel and interesting discovery and deserves to be published in Nature Communications. However, two important items remain to be addressed before publication. I don't think the manuscript needs to be reviewed again except by the editors.

1) Overall, the language needs considerable work for readability, clarity, and to remove typographical errors.

Response: We have used Springer Nature Language Editing service to improve the clarity and readability.

2) While the dynamical model is now better described, and the authors have made some attempt in their response to reviewers to give a physical explanation of their effect, this has still not made it into the main manuscript. It will take only a few lines but I think it is imperative for the authors to describe clearly the physical effect that makes vibration of chosen frequency and amplitude modulate the adhesion dramatically.

Response: Thank you very much for this valuable comment. As the first referee has the same concern, we just copy the response as:

Response: We have revised the **Abstract** and 5th paragraph to state the mechanism. Specifically, the revised **Abstract** are: "...Here we show a robust and predictable method ... perpendicular to the contact plane. We observed that the introduced micro-vibration leads to the contact interface experiencing rapid cracking and healing within one cycle. Based on the fact that the cracking and healing of an interface generally possess different rate dependences, we developed an analytic model to reveal the underlying mechanism, which intrinsically originates from adhesion hysteresis and dynamic instability. For typical PDMS-glass adhesion...".

In the 5th paragraph, the sentence (before Eq. (2)) " $w(a, v)$ is the effective adhesive work at the contact line and refers to Fig. S1 and Eq. (S1) for its dependence on v ($v \approx -\dot{a}$)" has been revised as " $w(a, v)$ is the effective adhesive work at the contact line and significantly and asymmetrically depends on the speed of the moving crack tip v ($v \approx -\dot{a}$). Specifically, as the crack propagation speed (v) increases, the adhesion work first increases exponentially, then falls sharply and finally rises rapidly again. As the healing speed ($|v|$) increases, the adhesion work rapidly approaches zero (refer to Fig. S1)".

REVIEWERS' COMMENTS:

Reviewer #1 (Remarks to the Author):

Overall, the edits are thorough and the main concerns of this reviewer have been addressed. Thank you for the hard work during this revision process.

Reviewer #2 (Remarks to the Author):

Although the authors have explained the underlying physics using text which is comprehensible, I would still recommend that a 2D-schematic would be useful for the readers to understand the concept. Other than this, I have no additional comments.

Reviewer #3 (Remarks to the Author):

I am satisfied with the second set of revisions. The manuscript can be accepted for publication. However, it does need to be edited carefully for typographical and grammatical errors.

RESPONSE TO REVIEWERS' COMMENTS:

Reviewer #1 (Remarks to the Author):

Overall, the edits are thorough and the main concerns of this reviewer have been addressed. Thank you for the hard work during this revision process.

Reviewer #2 (Remarks to the Author):

Although the authors have explained the underlying physics using text which is comprehensible, I would still recommend that a 2D-schematic would be useful for the readers to understand the concept. Other than this, I have no additional comments.

Reviewer #3 (Remarks to the Author):

I am satisfied with the second set of revisions. The manuscript can be accepted for publication. However, it does need to be edited carefully for typographical and grammatical errors.

Response: We thank the Reviewers for the approvals. Following reviewer #2's comment, we sketched a 2D-schematic about the concept (Fig. R1 or revised Fig. 4 in the manuscript). In addition, we used Springer Nature Language Editing service to improve the clarity and readability (Invoice# S9RXH4S4H).

Figure R1 The constitutive relation of effective adhesive work. The propagating and healing of an interface crack generally have different resistances, which results in the adhesion hysteresis. The inset shows a typical hysteresis loop under a cyclic load. For the situation of $v \leq v_c$, where v_c is the saturation velocity, the adhesion hysteresis can be utilized to increase the apparent adhesion work, hence enhance the interfacial adhesion strength. The experimentally determined α varies between 0.1 and 0.8, and the “N” shaped $w(v)$ curve is generally observed when $v > v_c$ ^{25-27,30}. More details can be found in **Supplementary Note 1**.